# Learning Monotonic Probabilities with a Generative Cost Model

**Yongxiang Tang** [1]  **Yanhua Cheng** [1]  **Xiaocheng Liu** [1]  **Chenchen Jiao** [1]  **Yanxiang Zeng** [1]  **Ning Luo** [1]
**Pengjia Yuan** [1]  **Xialong Liu** [1]  **Peng Jiang** [1]

## Abstract

In many machine learning tasks, it is often necessary for the relationship between input and output variables to be monotonic, including both strictly monotonic and implicitly monotonic relationships. Traditional methods for maintaining monotonicity mainly rely on construction or regularization techniques, whereas this paper shows that the issue of strict monotonic probability can be viewed as a partial order between an observable revenue variable and a latent cost variable. This perspective enables us to reformulate the monotonicity challenge into modeling the latent cost variable. To tackle this, we introduce a generative network for the latent cost variable, termed the Generative Cost Model (**GCM**), which inherently addresses the strict monotonic problem, and propose the Implicit Generative Cost Model (**IGCM**) to address the implicit monotonic problem. We further validate our approach with a numerical simulation of quantile regression and conduct multiple experiments on public datasets, showing that our method significantly outperforms existing monotonic modeling techniques. The code for our experiments can be found at https://github.com/tyxaaron/GCM.

## 1. Introduction

Many machine learning problems exhibit a monotonic relationship between inputs and outputs. These problems can be divided into two types: the first is strict monotonicity, where an increase in the input is assured to yield an increase in the output. For instance, the relationship between equipment usability and its age, or the relationship between auction winning rates and bidding prices. The second type is implicit monotonicity, often shown through correlations, such

as those between an individual's height and weight or between a company's stock price and its yearly earnings. For both types of monotonic problems, it's necessary to use a model that can predict the monotonic probability based on particular inputs. We call these input monotone variables as **revenue variables**, where higher revenue correlates with an increased probability of a more positive response.

Common deep learning methods to address the monotonicity problem can be broadly categorized into two types (Runje & Shankaranarayana, 2023): monotonic by **construction** and by **regularization**. The construction approach maintains strict monotonicity through customized structures in deep neural networks, such as nonconvex or nonconcave monotonic activation functions, positive weight matrices, and min-max structures (Sill, 1997). On the other hand, the regularization approach promotes monotonicity by designing specific loss functions (Sill & Abu-Mostafa, 1996); however, these approaches do not guarantee strict monotonicity.

Unlike conventional approaches, this paper introduces a **generative** framework to tackle the monotonicity problem. In our approach to estimating $p(y|\mathbf{x}, \mathbf{r})$, where $y \in \mathbb{R}$ is a response variable that maintains monotonicity with respect to the revenue variable $\mathbf{r}$ but not necessarily with $\mathbf{x}$, we employ a two-step process. (i) The regression task is converted into a classification task via variable substitution, reducing $y$ to a binary state (0 or 1). (ii) We proved that solving the monotonic binary classification problem is equivalent to identifying the latent **cost variable** $\mathbf{c}$, such that $\mathbf{c} \perp\!\!\!\perp \mathbf{r} \mid \mathbf{x}$ and $y \mid \{\mathbf{x}, \mathbf{r}\} \stackrel{d}{=} \mathbb{I}(\mathbf{c} \prec \mathbf{r}) \mid \{\mathbf{x}, \mathbf{r}\}$. Notably, $\stackrel{d}{=}$ indicates that two variables share an identical distribution, $\prec$ denotes the partial order in the vector space, and $\mathbb{I}$ is the indicator function. Consequently, we shift focus to modeling $\mathbf{c}$ and eliminate the need to design a strictly monotonic function, granting us increased flexibility in modeling $\mathbf{c}$.

This paper introduces a combined generative process for $\mathbf{x}$, $\mathbf{r}$, $\mathbf{c}$ and $y$. Although direct co-generation of $\mathbf{x}$, $\mathbf{r}$, and $\mathbf{c}$ via a single latent variable is not feasible since $\mathbf{c}$ must be conditionally independent of $\mathbf{r}$ given $\mathbf{x}$, i.e., $\mathbf{c} \perp\!\!\!\perp \mathbf{r} \mid \mathbf{x}$. To address this, we independently draw two different latent variables $\mathbf{z}$ and $\mathbf{w}$ from Gaussian priors and independently sample $\mathbf{x}$, $\mathbf{c}$, and $\mathbf{r}$ from $p(\mathbf{x}|\mathbf{z})$, $p(\mathbf{c}|\mathbf{z})$, and $p(\mathbf{r}|\mathbf{w}, \mathbf{x})$, ensuring $\mathbf{c} \perp\!\!\!\perp \mathbf{r} \mid \mathbf{x}$. Subsequently, $y$ is generated as $y = \mathbb{I}(\mathbf{c} \prec \mathbf{r})$.

[1]Kuaishou, Beijing, China. Correspondence to: Yongxiang Tang <tangyongxiang@kuaishou.com>, Yanhua Cheng <chengyanhua@kuaishou.com>.

*Proceedings of the $42^{nd}$ International Conference on Machine Learning*, Vancouver, Canada. PMLR 267, 2025. Copyright 2025 by the author(s).

For estimating the posterior $p(\mathbf{z}|\mathbf{x}, \mathbf{r}, \mathbf{y})$, we employ a variational model $q_\phi(\mathbf{z}|\mathbf{x})$, which is learned by optimizing the evidence lower bound (ELBO). This approach is termed the generative cost model (**GCM**).

Additionally, to capture implicit monotonicity, we alter the GCM generation procedure by integrating a latent kernel variable $\mathbf{k}$. This guarantees that both $\mathbf{r}$ and $\mathbf{y}$ exhibit monotonic behavior concerning $\mathbf{k}$. We refer to this refined method as the implicit generative cost model (**IGCM**). While this approach does not uphold strict monotonicity, it is better aligned with practical modeling applications.

In the last part, we conduct two types of experiments. First, we design a numerical simulation for quantile regression where the predicted r-th quantile increases monotonically with respect to the variable $r \in (0, 1)$. We evaluate the performance of traditional techniques versus our GCM approach, demonstrating that our method exhibits enhanced accuracy. To further evaluate the monotonic problem with a multivariate revenue variable $\mathbf{r}$, we conduct experiments on six publicly available datasets: the Adult dataset (Becker & Kohavi, 1996), the COMPAS dataset (Larson et al., 2016), the Diabetes dataset (Teboul), the Blog Feedback dataset (Buza, 2014), the Loan Defaulter dataset and the Auto MPG dataset (Quinlan, 1993). In most tasks, our model outperforms existing approaches. Further ablation studies and time complexity analysis are available in the appendix.

The main contributions of our paper are summarized as follows:

- We introduce a universal technique that reformulates the problem of monotonic probability into a modeling problem for latent cost variables, avoiding restrictions in conventional monotonic neural networks.

- We address the modeling of the cost variable using generative approaches, including the generative cost model (GCM) for strict monotonic problems and the implicit generative cost model (IGCM) for implicit monotonic problems.

- We evaluate our method through simulated quantile regression and machine learning tasks on multiple public datasets, demonstrating that our method consistently outperforms traditional monotonic models.

## 2. Background

**Definition 2.1** (Partial Order in Vector Space)**.** For vectors $\boldsymbol{v}_1$ and $\boldsymbol{v}_2$ in $\mathbb{R}^n$, $\boldsymbol{v}_1 \preceq \boldsymbol{v}_2$ if and only if $\boldsymbol{v}_1^{(k)} \leq \boldsymbol{v}_2^{(k)}$, for any $1 \leq k \leq n$.

This relationship is illustrated in Figure 1. Note that $\boldsymbol{v}_1 \preceq \boldsymbol{v}_2$ is equivalent to $\boldsymbol{v}_2 \succeq \boldsymbol{v}_1$.

**Definition 2.2** (Strict Order in Vector Space)**.** $\boldsymbol{v}_1 \prec \boldsymbol{v}_2$ if and only if $\boldsymbol{v}_1 \preceq \boldsymbol{v}_2$ and $\boldsymbol{v}_1 \neq \boldsymbol{v}_2$.

Note that $\boldsymbol{v}_1 \prec \boldsymbol{v}_2$ is equivalent to $\boldsymbol{v}_2 \succ \boldsymbol{v}_1$, but not equivalent to $\boldsymbol{v}_1 \not\succeq \boldsymbol{v}_2$.

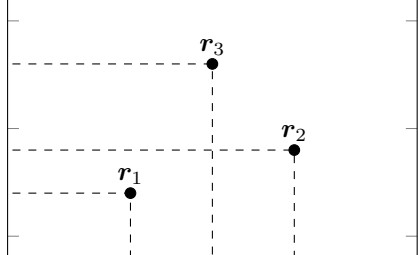

*Figure 1.* An example of partial order between vectors, where $\boldsymbol{r}_1 \prec \boldsymbol{r}_2$ and $\boldsymbol{r}_1 \prec \boldsymbol{r}_3$. The partial order between $\boldsymbol{r}_2$ and $\boldsymbol{r}_3$ is not determined.

**Definition 2.3** (First-Order Stochastic Dominance (Hadar & Russell, 1969))**.** For random variables $\mathbf{y}_1$ and $\mathbf{y}_2$ defined in $\mathbb{R}^n$, we say that $\mathbf{y}_2$ first-order stochastically dominates $\mathbf{y}_1$ (denoted $\mathbf{y}_1 \prec_{\text{r.v.}} \mathbf{y}_2$) if and only if $\Pr(\mathbf{y}_1 \succ \boldsymbol{t}) < \Pr(\mathbf{y}_2 \succ \boldsymbol{t})$ for any vector $\boldsymbol{t} \in \mathbb{R}^n$.

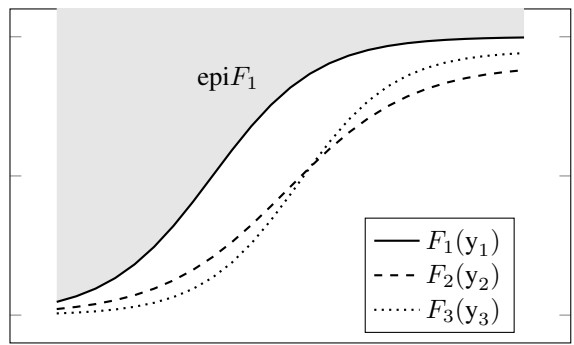

*Figure 2.* An illustration of the probability distribution functions for three random variables $\mathbf{y}_1$, $\mathbf{y}_2$, and $\mathbf{y}_3$, given that $\mathbf{y}_1 \prec_{\text{r.v.}} \mathbf{y}_2$ and $\mathbf{y}_1 \prec_{\text{r.v.}} \mathbf{y}_3$, is presented. It is apparent that $\text{epi}F_1$ is a subset of both $\text{epi}F_2$ and $\text{epi}F_3$. However, there is no containment relationship between the sets $\text{epi}F_2$ and $\text{epi}F_3$, indicating that the variables $\mathbf{y}_2$ and $\mathbf{y}_3$ cannot be directly compared.

It is significant to highlight that $[\mathbf{y}_1 \prec_{\text{r.v.}} \mathbf{y}_2]$ represents a statement, whereas $[\mathbf{y}_1 \prec \mathbf{y}_2]$ denotes an event, with its probability density expressed as $p(\mathbf{y}_1 \prec \mathbf{y}_2) = \iint_{\boldsymbol{y}_1 \prec \boldsymbol{y}_2} p(\mathbf{y}_1 = \boldsymbol{y}_1) p(\mathbf{y}_2 = \boldsymbol{y}_2) d\boldsymbol{y}_1 d\boldsymbol{y}_2$.

In the context of one-dimensional random variables, the relationship $\mathbf{y}_1 \prec_1 \mathbf{y}_2$ is equivalent to having $F_1(t) > F_2(t)$ for all real numbers $t$ (or $\text{epi}F_1 \subset \text{epi}F_2$), where $F_i$ denotes the cumulative distribution function (CDF) of the random variable $\mathbf{y}_i$, and $\text{epi}F_i$ indicates the epigraph of this CDF. This concept is illustrated in Figure 2.

For example, if $\mathbf{y} \sim \mathcal{N}(\mathbf{y}; \boldsymbol{\mu}, \boldsymbol{\Sigma})$, where the mean $\boldsymbol{\mu}$ is also

a random variable, then we find that $\mathbf{y}$ is monotonic with respect to $\boldsymbol{\mu}$. Similarly, if $y \sim \mathcal{B}ernoulli(\beta)$, then y is monotonic with respect to $\beta$. In these cases, $\boldsymbol{\mu}$ and $\beta$ are referred to as monotonic parameters.

**Definition 2.4** (Monotonic Conditional Probability). A conditional probability $p(\mathbf{y}|\mathbf{r})$ is defined as monotonic if and only if $\mathbf{y} \mid \{\mathbf{r} = \boldsymbol{r}_1\} \prec_{\text{r.v.}} \mathbf{y} \mid \{\mathbf{r} = \boldsymbol{r}_2\}$ for any pair of vectors $\boldsymbol{r}_1 \prec \boldsymbol{r}_2$.

Alternatively, $p(\mathbf{y}|\mathbf{r})$ is monotonic if and only if $\Pr(\mathbf{y} \succ \boldsymbol{t}|\mathbf{r} = \boldsymbol{r}_1) < \Pr(\mathbf{y} \succ \boldsymbol{t}|\mathbf{r} = \boldsymbol{r}_2)$ for any vector $\boldsymbol{t}$ and vectors $\boldsymbol{r}_1 \prec \boldsymbol{r}_2$. Specifically, in the Bernoulli case, $p(y|\mathbf{r})$ is considered monotonic if and only if $\Pr(y = 1|\mathbf{r} = \boldsymbol{r}_1) < \Pr(y = 1|\mathbf{r} = \boldsymbol{r}_2)$ for any pair of vectors $\boldsymbol{r}_1 \prec \boldsymbol{r}_2$. Within this paper, we refer to this relationship between $\mathbf{y}$ and $\mathbf{r}$ as: $\mathbf{y}$ being (conditionally) monotonic (increasing) with respect to $\mathbf{r}$. Throughout this paper, all discussions of monotonicity are assumed to be of the increasing type; for decreasing cases, the original variables can be substituted with their negatives.

## 3. Related Work

**Monotonic Modeling.** In numerous machine learning tasks, we hold the prior knowledge that the output should be monotonic with respect to certain input variables. A straightforward idea is to identify a monotonic function and optimize its parameters to approximate the desired monotonic output. It can be formulated as follows:

$$\text{minimize} \quad \mathcal{L}(\mathbf{y}, G_\theta(\mathbf{x}, \mathbf{r}))$$
$$\text{subject to} \quad \frac{\partial G_\theta(\mathbf{x}, \mathbf{r})}{\partial \mathbf{r}} \succ \mathbf{0}.$$

The Min-Max architecture (Sill, 1997) represents a foundational advancement in monotonic neural networks, utilizing a piecewise linear model to approximate monotonic target functions. This architecture maintains monotonicity through (i) positive weight matrices, (ii) monotonic activation functions, and (iii) a Min-Max structure.

Continuing in the direction of monotonic by construction, (Nolte et al., 2022) introduced the Lipschitz monotonic network, enhancing robustness through weight constraint. (Igel, 2024) proposed the smoothed min-max monotonic network by substituting the classic Min-Max structure with a smoothed log-sum-exp function, which prevents the neurons from becoming silent. (Runje & Shankaranarayana, 2023) developed the constrained monotonic neural network, refining the approximation of nonconvex functions through adjusted activation functions. Furthermore, (Kim & Lee, 2024) addressed the scalability limitations present in traditional monotonic models.

An alternative approach of monotonic modeling is through

regularization strategies, which can be formulated as:

$$\text{minimize} \quad \mathcal{L}(\mathbf{y}, G_\theta(\mathbf{x}, \mathbf{r})) + \mathcal{R}(G_\theta).$$

Here, the regularization $\mathcal{R}(G_\theta) > 0$ if $G_\theta$ is not monotonic at some points, while $\mathcal{R}(G_\theta) = 0$ if monotonicity is preserved. Crucially, the entire term $\mathcal{R}(G_\theta)$ is differentiable with respect to $\theta$, ensuring that the model adheres more closely to monotonicity. This direction includes monotonicity hints (Sill & Abu-Mostafa, 1996), which employ hint samples and pairwise loss to guide model learning. Certified monotonic neural networks (Liu et al., 2020) certify monotonicity by verifying the lower bound of the partial derivative of monotonic features. (Gupta et al., 2019) proposed a pointwise penalization method for negative gradients. Furthermore, counter example guided methods were introduced by (Sivaraman et al., 2020).

Additionally, lattice networks (Garcia & Gupta, 2009) address the monotonic problem via construction or regularization techniques; extensive works have been carried out in this area by (Milani Fard et al., 2016; You et al., 2017; Gupta et al., 2019; Yanagisawa et al., 2022).

Monotonicity is a significant concept in various machine learning domains. (Ben-David, 1995; Lee et al., 2003; van de Kamp et al., 2009; Chen & Guestrin, 2016) integrate monotonicity into tree-based models. The QMIX method (Rashid et al., 2020) incorporating monotonic value functions in multi-agent reinforcement learning. Additionally, (Lam et al., 2023) introduce a multi-class loss function utilizing the monotonicity of gradients of convex functions. Moreover, (Haldar et al., 2020) and (Xu et al., 2024) explore the application of monotonicity in online business.

**Generative Model via Variational Inference.** Variational inference (VI) (Peterson, 1987; Saul & Jordan, 1995) serves as a potent method for generative models, and has recently made considerable progress (Kingma, 2013; Rezende et al., 2014; Burda et al., 2016; Sohl-Dickstein et al., 2015; Ho et al., 2020; Song et al., 2021). VI simplifies the challenge of Bayesian inference by transforming it into a more feasible optimization problem, where latent variables are approximated within a chosen family of distributions. This is done by maximizing the evidence lower bound (ELBO) rather than the actual evidence.

Recently, there has been an emerging interest in examining generative models under specific conditions. For instance, significant contributions have appeared in text-to-image generation (Ramesh et al., 2021; 2022; Saharia et al., 2022; Rombach et al., 2022). Typically, these generative models initiate the generative process with predefined conditions (such as text, image, or video), generally represented as $p(\mathbf{x}, \mathbf{y}) = \mathbb{E}_{\mathbf{z} \sim p(\mathbf{z}|\mathbf{x})} p(\mathbf{x}) p(\mathbf{y}|\mathbf{x}, \mathbf{z})$, where x is the condition, z is the latent variable, and y is the target. Consequently, these models focus mainly on predicting the posterior $p(\mathbf{z}|\mathbf{x})$

and the conditional density $p(\mathrm{y}|\mathrm{x}, \mathrm{z})$. Since directly inferring the posterior $p(\mathrm{z}|\mathrm{x})$ is computationally challenging, the posterior is often approximated by the variational model $q(\mathrm{z}|\mathrm{x})$, optimized through the ELBO. This paper adopts this framework to build our generative cost model.

## 4. Method

### 4.1. Problem Formulation

In the case of a general monotonic problem involving $(\mathbf{x}, \mathbf{r}, \mathrm{y})$, where the output variable $\mathrm{y} \in \mathbb{R}$ is continuous, the model is structured as follows:

$$\mathrm{y} \mid \{\mathbf{x}, \mathbf{r}\} \sim \mathcal{F}(\mathrm{y}; G(\mathbf{x}, \mathbf{r})), \tag{1}$$

where $\mathcal{F}$ denotes the probability family for $\mathrm{y}$. The function $G$ is monotonic with respect to $\mathbf{r}$ and produces a monotonic parameter for $\mathcal{F}$. As a result, $\mathrm{y}$ preserves monotonicity with respect to $\mathbf{r}$. As an illustration, if $\mathcal{F}$ is a Gaussian distribution $\mathcal{N}\left(\mathrm{y}; \mu(\mathbf{x}, \mathbf{r}), \sigma(\mathbf{x})^2\right)$ with $G = \mu(\mathbf{x}, \mathbf{r})$ predicts its mean parameter, then $G$ must be a monotonic function of $\mathbf{r}$ to ensure that $y$ is monotonic with respect to $\mathbf{r}$.

This general monotonic probability problem can be transformed into the binary case by the following lemma.

**Lemma 4.1.** *For any random variable* $\mathrm{t} \in \mathbb{R}$ *defined over* $Dom(\mathrm{y})$, *the domain of* $\mathrm{y}$, *and that* $\mathrm{y} \perp\!\!\!\perp \mathrm{t} \mid \{\mathbf{x}, \mathbf{r}\}$. *Define* $\mathrm{y}^* = \mathbb{I}(\mathrm{y} > \mathrm{t}) \in \{0, 1\}$ *and* $\mathbf{r}^* = [-\mathrm{t}, \mathbf{r}]$. *Then* $\mathrm{y} \mid \{\mathbf{x}, \mathbf{r}\}$ *is a monotonic conditional probability with respect to* $\mathbf{r}$, *if and only if* $\mathrm{y}^* \mid \{\mathbf{x}, \mathbf{r}^*\}$ *is a monotonic conditional probability with respect to* $\mathbf{r}^*$.

The proof of this lemma is available in Appendix B.1. This lemma demonstrates the equivalence between the problems of $(\mathrm{y}^*, \mathbf{x}, \mathbf{r}^*)$ and $(\mathrm{y}, \mathbf{x}, \mathbf{r})$. As a result, the monotonic modeling task for the triplet $(\mathrm{y}, \mathbf{x}, \mathbf{r})$, where $\mathrm{y} \in \mathbb{R}$, can be simplified to the binary problem $(\mathrm{y}^*, \mathbf{x}, \mathbf{r}^*)$. Here, $\mathrm{y}^* \mid \{\mathbf{x}, \mathbf{r}^*\} \sim \mathcal{B}ernoulli(\mathrm{y}^*; G(\mathbf{x}, [-\mathrm{t}, \mathbf{r}]))$. Given that $\Pr(\mathrm{y} \leq s|\mathbf{x}, \mathbf{r}) = 1 - \Pr(\mathrm{y} > s|\mathbf{x}, \mathbf{r}) = 1 - G(\mathbf{x}, [-s, \mathbf{r}])$, the probability density function of $\mathrm{y}$ is given by:

$$p(\mathrm{y} = s|\mathbf{x}, \mathbf{r}) = -\frac{\partial G(\mathbf{x}, [-s, \mathbf{r}])}{\partial s}. \tag{2}$$

This equivalence illustrates the transformation of a general monotonic probability problem into a binary monotonic one. Additional information regarding this transformation is provided in Appendix D.3. As a result, our investigation now centers on the binary classification problem for $(\mathbf{x}, \mathbf{r}, \mathrm{y})$, where $\mathbf{x} \in \mathbb{R}^n$ represents the nonmonotonic variables, $\mathbf{r} \in \mathbb{R}^m$ signifies the monotonic variable, and $\mathrm{y} \in \{0, 1\}$ is the binary outcome that retains monotonicity with respect to $\mathbf{r}$. We refer to $\mathbf{r}$ as the **revenue variable** of $\mathrm{y}$. The reasoning is that when $\mathrm{y}$ is considered a decision variable, a profit-maximizing decision will favor higher values of $\mathbf{r}$, thereby ensuring the monotonicity between $\mathrm{y}$ and $\mathbf{r}$.

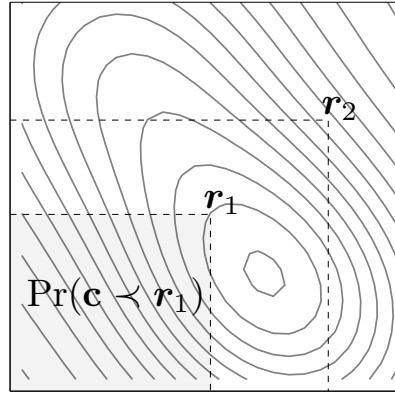

*Figure 3.* Within the density contour plot for the cost variable $\mathbf{c}$, the shaded area represents the event $\mathbf{c} \prec \mathbf{r}$. This event indicates that the probability of the cost variable $\mathbf{c}$ falling within this shaded region is expressed as $\Pr(\mathbf{c} \prec \mathbf{r})$. Thus, for any vectors $\boldsymbol{r}_1 \prec \boldsymbol{r}_2$, it follows that $\Pr(\mathbf{c} \prec \boldsymbol{r}_1) < \Pr(\mathbf{c} \prec \boldsymbol{r}_2)$.

The distribution of the Bernoulli variable $\mathrm{y}$ is defined by:

$$\mathrm{y} \mid \{\mathbf{x}, \mathbf{r}\} \sim \mathcal{B}ernoulli\left(\mathrm{y}; G(\mathbf{x}, \mathbf{r})\right). \tag{3}$$

According to Definition 2.4, the function $G$ is required to be monotonic with respect to $\mathbf{r}$. Learning the function $G$ poses significant difficulties, particularly when opting for a complex structure, such as a deep neural network. A considerable amount of prior research has concentrated on designing or learning an appropriate neural network for $G$. Our research, however, takes a different approach by introducing a latent cost variable, thereby eliminating the necessity of directly learning the function $G$.

### 4.2. The Cost Variable

In this section, we examine the binary problem. The conventional approach, as defined in Equation (3), involves identifying a strictly monotonic function $G(\mathbf{x}, \mathbf{r})$ with respect to $\mathbf{r}$. In this paper, instead of searching for a suitable function $G$, we introduce a random variable $\mathbf{c}$ to model $\mathrm{y}$. This random variable $\mathbf{c}$ fulfills the following conditions:

$$\mathbf{r} \perp\!\!\!\perp \mathbf{c} \mid \mathbf{x},$$

$$\mathrm{y} \mid \{\mathbf{x}, \mathbf{r}\} \stackrel{d}{=} \mathbb{I}(\mathbf{c} \prec \mathbf{r}) \mid \{\mathbf{x}, \mathbf{r}\},$$

where $\stackrel{d}{=}$ denotes that the two variables share the same distribution. The existence of the cost variable $\mathbf{c}$ is guaranteed by the following lemma.

**Lemma 4.2.** *Let* $\mathrm{y}$ *be a binary variable influenced by* $\mathbf{x}$ *and* $\mathbf{r}$, *with the constraint that* $\Pr(\mathrm{y} = 1|\mathbf{x}, \mathbf{r})$ *is continuous for* $\mathbf{r} \in D$, *where* $D$ *is bounded. Then* $\Pr(\mathrm{y} = 1|\mathbf{x}, \mathbf{r})$ *is monotonically increasing with respect to* $\mathbf{r}$ *if and only if there exists a random variable* $\mathbf{c}$ *in* $D$ *such that* $\mathbf{r} \perp\!\!\!\perp \mathbf{c} \mid \mathbf{x}$ *and* $\mathrm{y} \mid \{\mathbf{x}, \mathbf{r}\} \stackrel{d}{=} \mathbb{I}(\mathbf{c} \prec \mathbf{r}) \mid \{\mathbf{x}, \mathbf{r}\}$.

The proof of this lemma is available in Appendix B.2, and we also address the case of unbounded $\mathbf{r}$ in Appendix C. Here, we define $\mathbf{c}$ as the **cost variable**, with its intuitive interpretation akin to that of $\mathbf{r}$: it represents the resistance of a profit-maximizing decision. As shown in Figure 3, the likelihood of $y = 1$ corresponds to the probability that the revenue $\mathbf{r}$ domains the cost $\mathbf{c}$. This lemma implies that once the cost variable $\mathbf{c}$ is obtained, the initial probability $\Pr(y = 1|\mathbf{x}, \mathbf{r})$ varies monotonically with $\mathbf{r}$. Consequently, the task of identifying the monotonic function $G$ essentially transforms into determining the cost variable $\mathbf{c}$. However, since $\mathbf{c}$ is not directly observable, we must infer it from the observable variables $\mathbf{x}$, $\mathbf{r}$, and $y$, posing a significant unresolved challenge.

### 4.3. Generative Cost Model

In addressing the challenge of modeling the cost variable $\mathbf{c}$, the distribution of $\mathbf{c}$ can be complicated, making it challenging to select an appropriate distribution family. To bypass the necessity of picking a specific distribution family, we employ a generative approach capable of automatically approximating intricate distributions. A direct idea involves first sampling a latent variable $\mathbf{z}$, and subsequently generating $\mathbf{x}$, $\mathbf{r}$, and $\mathbf{c}$ independently, conditional on $\mathbf{z}$. However, this approach does not ensure $\mathbf{c} \perp\!\!\!\perp \mathbf{r} \mid \mathbf{x}$, as $\mathbf{z}$ still serves as a shared factor influencing both $\mathbf{c}$ and $\mathbf{r}$. To achieve $\mathbf{c} \perp\!\!\!\perp \mathbf{r} \mid \mathbf{x}$, we design a generative model using an additional latent variable $\mathbf{w}$ with the following process:

$$
\begin{aligned}
&\mathbf{z} \sim p_\theta(\mathbf{z}), \mathbf{w} \sim p_\theta(\mathbf{w}), \\
&\mathbf{x} \sim p_\theta(\mathbf{x}|\mathbf{z}), \mathbf{c} \sim p_\theta(\mathbf{c}|\mathbf{z}), \mathbf{r} \sim p_\theta(\mathbf{r}|\mathbf{w}, \mathbf{x}), \\
&y = \mathbb{I}(\mathbf{c} \prec \mathbf{r}).
\end{aligned}
\tag{4}
$$

Here, $p_\theta(\mathbf{z})$ and $p_\theta(\mathbf{w})$ are standard multivariate Gaussian distributions, while $p_\theta(\mathbf{x}|\mathbf{z})$, $p_\theta(\mathbf{c}|\mathbf{z})$, and $p_\theta(\mathbf{r}|\mathbf{w}, \mathbf{x})$ are conditional Gaussian distributions. This entire framework is referred to as the generative cost model (**GCM**) and is illustrated in Figure 4. It is evident from Figure 4 that $\mathbf{x}$ blocks the path between $\mathbf{r}$ and $\mathbf{c}$, and that $y$ is a collision node. Therefore, $\mathbf{r} \perp\!\!\!\perp \mathbf{c} \mid \mathbf{x}$ holds, ensuring the strict monotonicity of $y$ with respect to $\mathbf{r}$, as specified by Lemma 4.2.

For learning this generative model, we employ the variational inference method, utilizing the variational model $q_\phi(\mathbf{z}|\mathbf{x})$ to approximate the posterior $p_\theta(\mathbf{z}|\mathbf{r}, \mathbf{y})$. As a result, the variational bound is formulated as:

$$
\begin{aligned}
&\log p_\theta(\mathbf{x}, \mathbf{r}, \mathbf{y}) \\
&\geq \mathbb{E}_{\mathbf{z} \sim q_\phi} \log \frac{\Pr_\theta(\mathbf{c} \curlyvee_y \mathbf{r}|\mathbf{z}) p_\theta(\mathbf{z})}{q_\phi(\mathbf{z}|\mathbf{x})} + \log p_\theta(\mathbf{x}, \mathbf{r}|\mathbf{z}) \\
&\triangleq \text{ELBO}_{GCM}(\theta, \phi).
\end{aligned}
\tag{5}
$$

Here, $\curlyvee_y$ denotes $\prec$ if $y = 1$, and $\not\prec$ if $y = 0$ (note that $\not\prec$ is distinct from $\succeq$ in the vector space). The right hand

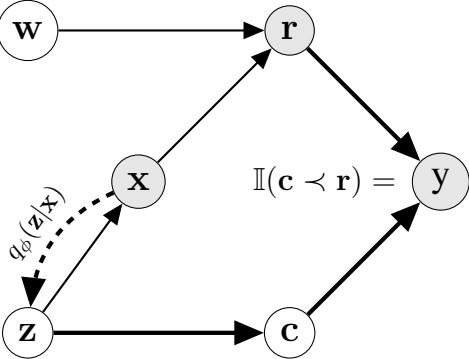

Figure 4. The probabilistic graphical model of the generative cost model ensures a monotonic conditional probability $p(y|\mathbf{x}, \mathbf{r})$ with respect to $\mathbf{r}$. Within this diagram, observable variables are depicted as gray nodes, whereas latent variables are shown as white nodes. Solid arrows illustrate the generative model $p_\theta$, and the dashed arrow indicates the variational model $q_\phi$. The thick arrows depict the estimator for $p(y|\mathbf{x}, \mathbf{r})$.

side of this inequality corresponds to the evidence lower bound (ELBO) of our model, with equality occurring only if $D_{KL}[q_\phi(\mathbf{z}|\mathbf{x})\|p(\mathbf{z}|\mathbf{x}, \mathbf{r}, \mathbf{y})] = 0$. The detailed derivation can be found in Appendix D.1. Comprehensive details on computing $\Pr_\theta(\mathbf{c} \curlyvee_y \mathbf{r}|\mathbf{z}, \mathbf{r})$ can be found in Appendix D.2.

In the course of optimizing the ELBO, our model employs the reparameterization trick (Kingma, 2013) as $\boldsymbol{z}^{(n)} = \boldsymbol{\mu}(\mathbf{x}) + \boldsymbol{\sigma}(\mathbf{x}) \odot \boldsymbol{\epsilon}^{(n)}$, where $\boldsymbol{\epsilon}^{(n)}$ is sampled from the standard normal distribution $\mathcal{N}(\mathbf{0}, \mathbf{E})$. Following the IWAE (Burda et al., 2016) framework, the final loss function of GCM is given by:

$$
\begin{aligned}
&\mathcal{L}_{GCM}(\theta, \phi; \boldsymbol{x}, \boldsymbol{r}, y) \\
&= -\log \frac{1}{N} \sum_{n=1}^{N} \frac{\Pr_\theta\left(\mathbf{c} \curlyvee_{y=y} \boldsymbol{r}|\mathbf{z} = \boldsymbol{z}^{(n)}\right) p_\theta\left(\mathbf{z} = \boldsymbol{z}^{(n)}\right)}{q_\phi\left(\mathbf{z} = \boldsymbol{z}^{(n)}|\mathbf{x} = \boldsymbol{x}\right)}.
\end{aligned}
\tag{6}
$$

Here, we drop the term $p_\theta(\mathbf{x}, \mathbf{r}|\mathbf{z})$ in Equation (5), since we do not need it for predicting $y$. The sampling number $N$ influences the loss function, with a detailed ablation study on $N$ provided in Appendix F. In scenarios where the output variable $y$ is continuous, we additionally formulate the corresponding loss function, which is elaborated in Appendix D.3.

During inference, the aim is to estimate $y$ while given $\mathbf{x}$ and $\mathbf{r}$. We leverage the approximation $p_\theta(\mathbf{z}|\mathbf{x}, \mathbf{r}, \mathbf{y}) \simeq q_\phi(\mathbf{z}|\mathbf{x})$ to acquire:

$$
\begin{aligned}
&\Pr_\theta(y = 1|\mathbf{x} = \boldsymbol{x}, \mathbf{r} = \boldsymbol{r}) \\
&= \mathbb{E}_{\mathbf{z} \sim p_\theta(\mathbf{z}|\mathbf{x}=\boldsymbol{x}, \mathbf{r}=\boldsymbol{r}, y=1)} \Pr_\theta(y = 1|\mathbf{z}, \mathbf{r} = \boldsymbol{r}) \\
&\simeq \frac{1}{N} \sum_{n=1}^{N} \Pr_\theta\left(\mathbf{c} \prec \boldsymbol{r}|\mathbf{z} = \boldsymbol{z}^{(n)}\right),
\end{aligned}
$$

where $\boldsymbol{z}^{(n)}$ is a random sample from $q_\phi(\mathbf{z}|\mathbf{x} = \boldsymbol{x})$. This estimation is strictly monotonic with respect to $\boldsymbol{r}$, since $\Pr_\theta\left(\mathbf{c} \prec \boldsymbol{r}|\mathbf{z} = \boldsymbol{z}^{(n)}\right)$ is a strict monotonic function of $\boldsymbol{r}$. Consequently, we have established strict monotonicity between $\boldsymbol{r}$ and y without the necessity of developing a monotonic function $G(\boldsymbol{x}, \boldsymbol{r})$, thereby reducing the complexity involved in designing a strict monotonic neural network.

### 4.4. Implicit Generative Cost Model

While we have effectively established the generative cost model to address the strict monotonic problem, a challenge remains in that not all monotonic relationships indicate causation; some merely reflect correlation. For instance, both a child's height and weight increase monotonically with age, which serves as the underlying monotonic factor. In considering the monotonic relationship between the output variable y and the revenue variable $\mathbf{r}$, our goal is to identify the latent variable $\mathbf{k}$, with which both y and $\mathbf{r}$ exhibit monotonicity. This model does not guarantee strict monotonicity of $p(\mathbf{y}|\mathbf{x}, \mathbf{r})$; however, it introduces a monotonic correlation between y and $\mathbf{r}$, owing to the strict monotonicity of $p(\mathbf{y}|\mathbf{x}, \mathbf{k})$ and $p(\mathbf{r}|\mathbf{k})$. We designate $\mathbf{k}$ as the kernel revenue variable.

The generative framework is specified as follows:

$$\mathbf{z} \sim p_\theta(\mathbf{z}), \mathbf{k} \sim p_\theta(\mathbf{k}),$$
$$\mathbf{x} \sim p_\theta(\mathbf{x}|\mathbf{z}, \mathbf{k}), \mathbf{c} \sim p_\theta(\mathbf{c}|\mathbf{z}), \Delta\mathbf{r} \sim p_\theta(\Delta\mathbf{r}|\mathbf{x}),$$
$$\mathbf{r} = \Delta\mathbf{r} + \boldsymbol{W}\mathbf{k}, \boldsymbol{W} \succ \mathbf{0},$$
$$\mathbf{y} = \mathbb{I}(\mathbf{c} \prec \mathbf{k}).$$

The assurance of a monotonic relationship between $\mathbf{k}$ and $\mathbf{r}$ is given by the positive projection matrix $\boldsymbol{W}$, while the monotonic relationship between y and $\mathbf{k}$ is guaranteed by $\mathbf{y} = \mathbb{I}(\mathbf{c} \prec \mathbf{k})$ and the independence condition $\mathbf{c} \perp\!\!\!\perp \mathbf{k} \mid \mathbf{x}$. This model is referred to as the implicit generative cost model (**IGCM**), as shown in Figure 5.

To learn the IGCM, we propose an additional variational model $q_\psi(\mathbf{k}|\mathbf{x}, \mathbf{r})$ to estimate the posterior distribution $p_\theta(\mathbf{k}|\mathbf{x}, \mathbf{r}, \mathbf{y})$. Close to Equation (5), the ELBO of IGCM can be derived as:

$$\text{ELBO}_{IGCM}(\theta, \phi, \psi)$$
$$= \mathbb{E}_{\mathbf{z} \sim q_\phi, \mathbf{k} \sim q_\psi} \log \Pr_\theta(\mathbf{c} \curlyvee_\mathbf{y} \mathbf{k}|\mathbf{z})p_\theta(\mathbf{x}|\mathbf{z})p_\theta(\mathbf{r}|\mathbf{k})$$
$$- \mathbb{E}_{\mathbf{z} \sim q_\phi} \log \frac{p_\theta(\mathbf{z})}{q_\phi(\mathbf{z}|\mathbf{x})} - \mathbb{E}_{\mathbf{k} \sim q_\psi} \log \frac{p_\theta(\mathbf{k})}{q_\psi(\mathbf{k}|\mathbf{x}, \mathbf{r})}.$$

Similarly to Equation (6), we drop the term $p_\theta(\mathbf{x}|\mathbf{z})$ in $\text{ELBO}_{IGCM}(\theta, \phi, \psi)$, while keeping the term $p_\theta(\mathbf{r}|\mathbf{k})$ to capture the latent monotonicity between $\mathbf{r}$ and $\mathbf{k}$. The corre-

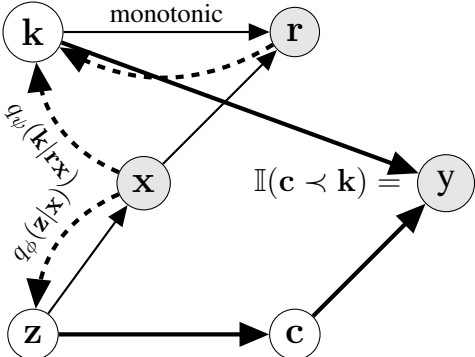

*Figure 5.* The probability graphical model for the implicit generative cost model (IGCM), which doses not ensure strict monotonic between $\mathbf{r}$ and y, but introduces monotonicity from $\mathbf{k}$ to $\mathbf{r}$ and from $\mathbf{k}$ to y. The thick arrows represent the path for the prediction of $p(\mathbf{y}|\mathbf{x}, \mathbf{r})$.

sponding loss function becomes:

$$\mathcal{L}_{IGCM}(\theta, \phi, \psi; \boldsymbol{x}, \boldsymbol{r}, y)$$
$$= -\log \frac{1}{N} \sum_{n=1}^{N} \mathbb{E}_{\mathbf{k} \sim q_\psi} \Pr_\theta\left(\mathbf{c} \curlyvee_{\mathbf{y}=y} \mathbf{k}|\mathbf{z} = \boldsymbol{z}^{(n)}\right)$$
$$\cdot \frac{p_\theta\left(\mathbf{z} = \boldsymbol{z}^{(n)}\right)}{q_\phi\left(\mathbf{z} = \boldsymbol{z}^{(n)}|\mathbf{x} = \boldsymbol{x}\right)}$$
$$- \log \frac{1}{M} \sum_{m=1}^{M} p_\theta(\mathbf{r} = \boldsymbol{r}|\mathbf{k} = \boldsymbol{k}^{(m)})$$
$$+ D_{KL}\left[q_\psi(\mathbf{k}|\mathbf{x} = \boldsymbol{x}, \mathbf{r} = \boldsymbol{r})\|p_\theta(\mathbf{k})\right].$$

Since the conditional distributions $p_\theta(\mathbf{c}|\mathbf{z})$, $p_\theta(\mathbf{r}|\mathbf{k})$, and $q_\psi(\mathbf{k}|\mathbf{x}, \mathbf{r})$ are all Gaussian, the expressions $\mathbb{E}_{\mathbf{k} \sim q_\psi} \Pr_\theta(\mathbf{c} \curlyvee_{\mathbf{y}=y} \mathbf{k}|\mathbf{z} = \boldsymbol{z}^{(n)})$, $p_\theta(\mathbf{r} = \boldsymbol{r}|\mathbf{k} = \boldsymbol{k}^{(m)})$ and $D_{KL}\left[q_\psi(\mathbf{k}|\mathbf{x}, \mathbf{r})\|p_\theta(\mathbf{k})\right]$ can be evaluated in closed form. The vectors $\boldsymbol{z}^{(n)}$ and $\boldsymbol{k}^{(m)}$ represent samples from their respective variational distributions $q_\phi(\mathbf{z}|\mathbf{x})$ and $q_\psi(\mathbf{k}|\mathbf{x})$. Consequently, the reparameterization trick can be employed to optimize this objective effectively. The estimator for y within the IGCM framework is ultimately represented as:

$$\Pr_\theta(\mathbf{y} = 1|\mathbf{x} = \boldsymbol{x}, \mathbf{r} = \boldsymbol{r})$$
$$= \mathbb{E}_{\mathbf{z}, \mathbf{k} \sim p_\theta(\mathbf{z}, \mathbf{k}|\mathbf{x}=\boldsymbol{x}, \mathbf{r}=\boldsymbol{r}, \mathbf{y}=1)} \Pr_\theta(\mathbf{y} = 1|\mathbf{z}, \mathbf{k})$$
$$\simeq \frac{1}{N} \sum_{n=1}^{N} \mathbb{E}_{\mathbf{k} \sim q_\psi} \Pr_\theta\left(\mathbf{c} \prec \mathbf{k}|\mathbf{z} = \boldsymbol{z}^{(n)}\right),$$

in which $\boldsymbol{z}^{(n)}$ are sampled from $q_\phi(\mathbf{z}|\mathbf{x})$, and the variable $\mathbf{k}$ is estimated using the variational distribution $q_\psi(\mathbf{k}|\mathbf{x}, \mathbf{r})$. In contrast to the GCM, the IGCM estimator does not maintain a strict monotonic property.

*Table 1.* MAE (with 95% confidence interval) of the quantile regression experiment.

| Method | MAE | | | | |
| --- | --- | --- | --- | --- | --- |
| | r=0.1 | r=0.3 | r=0.5 | r=0.7 | r=0.9 |
| PosNN | 0.2103 ±0.0464 | 0.1174 ±0.0327 | 0.0900 ±0.0140 | 0.1173 ±0.0291 | 0.1898 ±0.0332 |
| MM | 0.2055 ±0.0623 | 0.1202 ±0.0339 | 0.0819 ±0.0236 | 0.1236 ±0.0376 | 0.1781 ±0.0551 |
| SMM | 0.2499 ±0.0799 | 0.1310 ±0.0410 | 0.0846 ±0.0274 | 0.1420 ±0.0378 | 0.2144 ±0.0620 |
| CMNN | 0.1575 ±0.0349 | 0.1096 ±0.0253 | 0.0799 ±0.0287 | 0.1165 ±0.0348 | 0.1685 ±0.0428 |
| SMNN | 0.2184 ±0.0486 | 0.1132 ±0.0319 | 0.0870 ±0.0143 | 0.1140 ±0.0313 | 0.1777 ±0.0622 |
| Hint | 0.1536 ±0.0241 | 0.1166 ±0.0153 | 0.1091 ±0.0181 | 0.1280 ±0.0211 | 0.1379 ±0.0164 |
| PWL | 0.2083 ±0.0217 | 0.1587 ±0.0138 | 0.1436 ±0.0131 | 0.1608 ±0.0162 | 0.1720 ±0.0187 |
| GCM | **0.0961** ±0.0261 | **0.0817** ±0.0318 | **0.0742** ±0.0233 | **0.0792** ±0.0159 | **0.0899** ±0.0155 |

## 5. Experiment

### 5.1. Experiment by Simulation

Quantile regression is a common problem in statistics, aiming to estimate the r-th quantile of y given x, based on observations of both x and y. The r-th quantile, represented as $Q_{y|x}(r)$, is defined by $Q_{y|x}(r) = F_{y|x}^{-1}(r)$, where $F_{y|x}$ is the conditional cumulative distribution function of y given x. Due to the monotonic nature of $F$, its inverse, $Q_{y|x}(r)$, is also strictly monotonic with respect to r. The standard objective of linear quantile regression (Koenker, 2005) is formulated as:

$$\hat{\beta}_r = \arg\min_{\beta_r} \sum_i \rho_r \left( y^{(i)} - \hat{y}_r^{(i)} \right),$$

$$\text{where } \rho_r(\Delta) = r\Delta_+ + (1-r)(-\Delta)_+.$$

Here, $\Delta_+ = \Delta$ if $\Delta > 0$, otherwise $\Delta_+ = 0$. $(x^{(i)}, y^{(i)})$ is the $i$-th example pair and $\hat{y}_r^{(i)} = x^{(i)}\beta_r$ denotes the linear prediction of the quantile $Q_{y|x=x^{(i)}}(r)$, with $\beta_r$ as its parameter. For nonlinear relationships of y given x, neural networks can be leveraged to automatically capture these dynamics. Additionally, by integrating r into the network, it is possible to predict the r-th quantile of y given x using $\hat{y}_r = \text{DNN}_\theta(x, r)$, for any $r \in (0, 1)$. This can also be represented in a generative format:

$$y_r \sim p_\theta(y_r|x, r),$$
$$\hat{y}_r = \mathbb{E} y_r.$$

However, this problem diverges from the original monotonic modeling, since the variable r is unobservable, preventing the direct implementation of the GCM method. To address this, we reformulate the quantile regression problem as follows:

$$\text{sample } r \sim \mathcal{U}([0, 1]),$$
$$\text{minimize } \mathbb{E}_{y_r \sim p_\theta(y_r|x=x, r=r)} \rho_r(y - y_r), \quad (7)$$

which becomes an optimizable problem.

This paper evaluates various traditional methods alongside our generative cost model (GCM). The strictly monotonic characteristics of this context exclude the IGCM method from consideration. All methods share an underlying three-layer perceptron network architecture with $\tanh$ activation functions. For training, network parameters are optimized using the stochastic gradient descent algorithm. The methods being compared include: (i) the positive weighted neural network (PosNN), which employs solely positive weight matrices (refer to Appendix E); (ii) the Min-Max network (MM) (Sill, 1997); (iii) the smooth Min-Max network (SMM) (Igel, 2024); (iv) the constrained monotonic network (CMNN) (Runje & Shankaranarayana, 2023); (v) the scalable monotonic network (SMNN) (Kim & Lee, 2024); (vi) the monotonicity hint model (Hint) (Sill & Abu-Mostafa, 1996); (vii) the pointwise loss method (PWL) (Gupta et al., 2019). It should be noted that both the Hint and PWL techniques are weakly monotonic, aiming to enhance monotonicity without ensuring it. Training data are generated using a simulation configured as follows:

$$x \sim \mathcal{U}(-1.5, 1.5),$$
$$\epsilon \sim \mathcal{N}(0, 1),$$
$$y = 0.3\sin(2(x + 0.8)) + 0.4\sin(3(x - 1.3))$$
$$+ 0.3\sin(5x) + 0.2(0.8x^2 + 0.6)\epsilon. \quad (8)$$

For each example $(x, y) = (x^{(i)}, y^{(i)})$, we additionally sample $r^{(i)} \sim \mathcal{U}([0, 1])$ and optimize our models with Equation (7). It is important to note that the sampling of $r^{(i)}$ operates independently of $\epsilon$. Our training procedure employs a batch size of 20 over 5,000 iterations, yielding a total of 100,000 training samples. The testing phase involves 1,000 samples per quantile, with $r \in \{0.1, 0.3, 0.5, 0.7, 0.9\}$. Table 1 provides the mean absolute error (MAE) results for all methods involved in the quantile regression analysis. We conduct the experiment 10 times, varying the random seeds each time, and present the results with 95% confidence intervals. According to the results, GCM surpasses all competing methods.

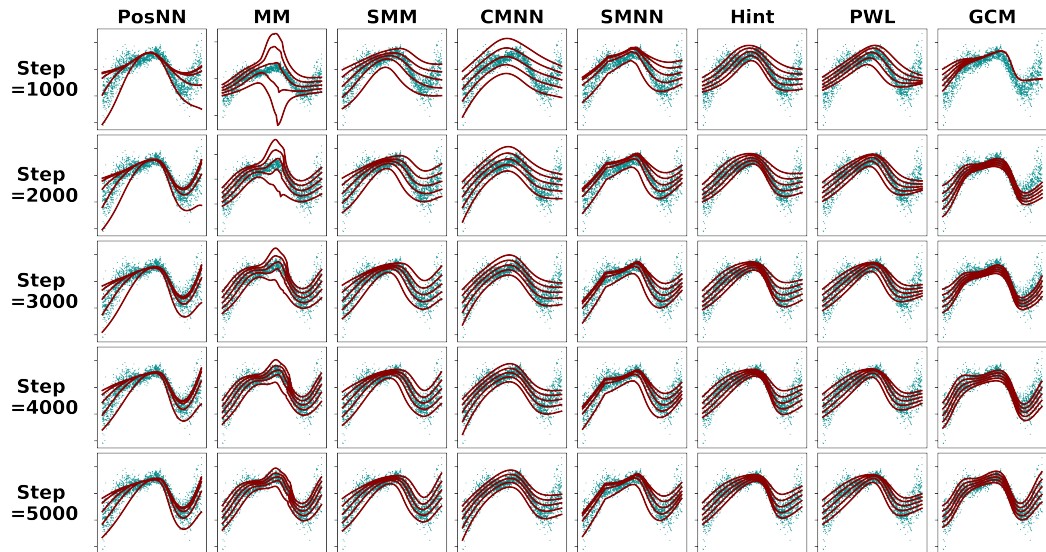

*Figure 6.* Validation of quantile regression accuracy. The background scatter points represent the true examples sampled from the distribution specified in Equation (8), while the red curves depict the estimated quantile curves obtained by each model at various training stages ranging from 0 to 5, 000.

We define the $r$-th quantile curve as $\Gamma_r = \{(x, y_r)\}$; thus, for any $r_2 > r_1$, the quantile's monotonic property ensures that $\Gamma_{r_2}$ is positioned above $\Gamma_{r_1}$. Figure 6 presents the quantile curve prediction outcomes across the training phases, where each column denotes a distinct model approach and each row corresponds to a training stage. Every panel depicts five predicted quantile curves $\Gamma_r$ for $r \in \{0.1, 0.3, 0.5, 0.7, 0.9\}$ illustrated as red curves, while the training data $(x_i, y_i)$ are represented as underlying scatter points, demonstrating the prediction efficacy of each model at different training stages. Ideally, the predicted quantile curves should closely approximate the true quantile curve, defined as:

$$
\begin{aligned}
y_r =\ & 0.3 \sin(2(x + 0.8)) + 0.4 \sin(3(x - 1.3)) \\
& + 0.3 \sin(5x) + 0.2(0.8x^2 + 0.6)\Phi^{-1}(r),
\end{aligned}
$$

where $\Phi$ denotes the CDF of the standard normal distribution. Conceptually, the predicted quantile curves ought to be distinct from each other, with the upper curve $\Gamma_{0.9}$ and the lower curve $\Gamma_{0.1}$ positioned towards the edges of the scatter plot, with limited outliers.

Inspection of Figure 6 indicates that: (i) the quantile curves for PosNN, MM, SMM, and SMNN, which are primarily built using monotonic construction methods, tend to cluster too closely in certain regions; (ii) CMNN, Hint, and PWL, which mainly employ monotonic regularization techniques, have a surplus of outliers beyond $\Gamma_{0.1}$ and $\Gamma_{0.9}$; and (iii) GCM demonstrates noticeable separations between the quantile curves and the fewest instances of outliers. Ultimately, GCM yields the most precise quantile curve predictions in contrast to traditional approaches.

## 5.2. Experiments on Public Datasets

To further evaluate the GCM and IGCM methods with multidimensional revenue variables, we conduct experiments using six public datasets, which include the Adult dataset (Becker & Kohavi, 1996), the COMPAS dataset (Larson et al., 2016), the Diabetes dataset (Teboul), the Blog Feedback dataset (Buza, 2014), the Loan Defaulter dataset and the Auto MPG dateset (Quinlan, 1993). The main statistics of each dataset are presented in Table 2.

*Table 2.* Details of the datasets.

| dataset | examples | x dim. | r dim. | target |
|---|---|---|---|---|
| Adult | 48,842 | 33 | 4 | classification |
| COMPAS | 6,172 | 9 | 4 | classification |
| Diabetes | 253,680 | 105 | 4 | classification |
| Blog Feedback | 54,270 | 268 | 8 | regression |
| Loan Defaulter | 488,909 | 23 | 5 | classification |
| Auto MPG | 392 | 4 | 3 | regression |

The models we compare are identical to those presented in Section 5.1, while the evaluation metrics are switched to root mean squared error (RMSE), classification accuracy (ACC) and area under the curve (AUC). For regression tasks, we apply the RMSE metric, while for classification tasks with a positive sample ratio between $[0.3, 0.7]$, we utilize the ACC metric; otherwise, the AUC metric is employed. The datasets are divided into training and testing sets in a $4 : 1$ ratio. We adhere to the data preprocessing methods specified by (Runje & Shankaranarayana, 2023) for the COMPAS, Blog Feedback, Loan Defaulter, and Auto MPG datasets.

*Table 3.* Experimental results on public datasets.

| Method | Adult AUC↑ | COMPAS ACC↑ | Diabetes AUC↑ | Blog Feedback RMSE↓ | Loan Defaulter ACC↑ | Auto MPG RMSE↓ |
|---|---|---|---|---|---|---|
| PosNN | 0.7597 ±0.0016 | 0.6921 ±0.0013 | 0.8254 ±0.0002 | 0.1580 ±0.0017 | 0.6519 ±0.0003 | 3.1473 ±0.1417 |
| MM | 0.7791 ±0.0010 | **0.6976** ±0.0010 | 0.8263 ±0.0003 | 0.1591 ±0.0005 | 0.6526 ±0.0003 | 3.4517 ±0.1311 |
| SMM | 0.7819 ±0.0012 | 0.6944 ±0.0016 | 0.8263 ±0.0001 | 0.1579 ±0.0006 | 0.6527 ±0.0002 | 3.3534 ±0.0923 |
| CMNN | 0.7725 ±0.0009 | 0.6918 ±0.0012 | 0.8261 ±0.0003 | 0.1685 ±0.0030 | 0.6525 ±0.0002 | 3.2813 ±0.1099 |
| SMNN | 0.7729 ±0.0033 | 0.6935 ±0.0015 | 0.8245 ±0.0002 | 0.1776 ±0.0097 | 0.6527 ±0.0001 | 3.3640 ±0.1649 |
| Hint* | 0.7509 ±0.0009 | 0.6723 ±0.0013 | 0.8184 ±0.0002 | 0.1590 ±0.0011 | 0.6477 ±0.0003 | 3.3872 ±0.1689 |
| PWL* | 0.7661 ±0.0006 | 0.6916 ±0.0014 | 0.8251 ±0.0003 | 0.1606 ±0.0031 | 0.6522 ±0.0003 | 3.4017 ±0.1302 |
| GCM | 0.7844 ±0.0025 | 0.6945 ±0.0011 | 0.8263 ±0.0004 | 0.1584 ±0.0004 | **0.6528** ±0.0004 | 3.2240 ±0.0801 |
| IGCM* | **0.7891** ±0.0018 | 0.6934 ±0.0012 | **0.8285** ±0.0002 | **0.1569** ±0.0005 | **0.6528** ±0.0004 | **3.1000** ±0.1268 |

∗: Non-strict monotonic model.

The hyperparameter configurations for GCM and IGCM can be found in Appendix E. Each experiment is repeated 10 times with different random seeds, and the results are reported with 95% confidence intervals, as shown in Table 3. To better understand the experimental results, we analyze the test results in three aspects: (i) comparison between GCM and IGCM, (ii) comparison between strict monotonic methods, and (iii) comparison between non-strict monotonic methods.

**Comparison between GCM and IGCM.** Our analysis reveals that within public datasets, the IGCM outperforms the GCM. This improved performance is attributed to the fact that real-world scenarios do not always adhere to strict monotonic relationships; for instance, while BMI is correlated with diabetes, it is not necessarily causative. Typically, the selection of monotonic features is informed by experience or statistical analysis. Under these circumstances, implicit monotonic assumptions tend to be more applicable. Our experimental data support this assertion, as IGCM demonstrates superior performance to GCM in five of the six datasets.

**Comparison between strict monotonic methods.** The strict monotonic methods evaluated include PosNN, MM, SMM, CMNN, SMNN, and the proposed GCM. Our evaluation indicates that the basic PosNN performs inadequately on most datasets, implying that simply implementing a positively weighted neural network is insufficient for handling monotonic problems. Notably, the traditional MM method, along with its successor SMM, demonstrates considerable effectiveness, indicating that the fundamental Min-Max architecture successfully optimizes the monotonic network. Recent advancements with CMNN and SMNN introduce novel monotonic network structures that surpass MM in several tests. Our proposed GCM method consistently achieves strong performance across all datasets, suggesting that reframing the monotonic problem as a generative one yields

improved outcomes compared to merely modifying the network structure.

**Comparison between non-strict monotonic methods.** Hint, PWL, and IGCM are non-strict monotonic approaches, with Hint and PWL serving as regularization techniques. In contrast, our newly developed IGCM tackles the non-strict issue via an innovative implicit monotonic modeling. This involves addressing two strict monotonic challenges: $p(\mathbf{r}|\mathbf{k})$ and $p(\mathbf{y}|\mathbf{k})$, thereby creating a non-strict monotonic relationship between y and r. This new technique offers an innovative probabilistic perspective for non-strict monotonic modeling. Experimental findings indicate enhancements when compared to conventional non-strict monotonic methods.

In conclusion, our GCM and IGCM methods deliver superior performance in most tasks, highlighting the efficacy of our generative approach. Further analysis of time complexity is provided in Appendix G and ablation studies are presented in Appendix F.

## 6. Conclusion

This paper introduces a novel generative approach to monotonic modeling. We reformulate the monotonicity problem by incorporating a latent cost variable **c**. Additionally, we propose two generative methods: GCM, which targets the strict monotonic problem, and IGCM, which focuses on the implicit monotonic problem. Our experimental findings indicate that GCM and IGCM successfully tackle the monotonicity challenge and substantially surpass traditional methods across multiple tasks.

## Impact Statement

This paper presents work whose goal is to advance the field of Machine Learning. There are many potential societal

consequences of our work, none which we feel must be specifically highlighted here.

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

# A. Notations

*Table 4.* Notations.

| notation type | examples |
|---|---|
| scalar | $x, r, y, c$ |
| vector | $\boldsymbol{x}, \boldsymbol{r}, \boldsymbol{y}, \boldsymbol{c}$ |
| matrix | $\boldsymbol{W}$ |
| random variable | x, r, y, c |
| random vector | $\mathbf{x}, \mathbf{r}, \mathbf{y}, \mathbf{c}$ |
| model parameter | $\theta, \phi, \psi$ |
| function | $F, G$ |
| distribution | $\mathcal{F}, \mathcal{N}$ |
| density function | $p, q$ |

# B. Proofs

## B.1. Proof of Lemma 1

*Proof.* Since t $\perp\!\!\!\perp$ y $\mid \{\mathbf{x}, \mathbf{r}\}$, for any $\boldsymbol{r}_0^* = [t_0, \boldsymbol{r}_0]$, it always holds that

$$
\begin{aligned}
&\Pr(\mathrm{y}^* = 1 | \mathbf{x}, \mathbf{r}^* = \boldsymbol{r}_0^*) \\
&= \Pr(\mathrm{y} > \mathrm{t} | \mathrm{t} = t_0, \mathbf{x}, \mathbf{r} = \boldsymbol{r}_0) \\
&= \Pr(\mathrm{y} > t_0 | \mathrm{t} = t_0, \mathbf{x}, \mathbf{r} = \boldsymbol{r}_0) \\
&= \Pr(\mathrm{y} > t_0 | \mathbf{x}, \mathbf{r} = \boldsymbol{r}_0).
\end{aligned}
\tag{9}
$$

If y $\mid \{\mathbf{x}, \mathbf{r}\}$ is a monotonic conditional probability with respect to $\mathbf{r}$. For any $\boldsymbol{r}_1^* \prec \boldsymbol{r}_2^*$, where $\boldsymbol{r}_i^* = [-t_i, \boldsymbol{r}_i]$, it follows that $\boldsymbol{r}_1 \preceq \boldsymbol{r}_2$ and $t_1 \geq t_2$. By employing Definition 2.4, the following inequality holds:

$$
\begin{aligned}
&\Pr(\mathrm{y}^* = 1 | \mathbf{x}, \mathbf{r}^* = \boldsymbol{r}_1^*) \\
&= \Pr(\mathrm{y} > t_1 | \mathbf{x}, \mathbf{r} = \boldsymbol{r}_1) && \text{(by Equation (9))} \\
&\leq \Pr(\mathrm{y} > t_1 | \mathbf{x}, \mathbf{r} = \boldsymbol{r}_2) && \text{(by } \boldsymbol{r}_1 \preceq \boldsymbol{r}_2) \\
&\leq \Pr(\mathrm{y} > t_2 | \mathbf{x}, \mathbf{r} = \boldsymbol{r}_2) && \text{(by } t_1 \geq t_2) \\
&= \Pr(\mathrm{y}^* = 1 | \mathbf{x}, \mathbf{r}^* = \boldsymbol{r}_2^*) && \text{(by Equation (9)).}
\end{aligned}
$$

Equality occurs if $t_1 = t_2$ and $\boldsymbol{r}_1 = \boldsymbol{r}_2$, which is inconsistent with $\boldsymbol{r}_1^* \prec \boldsymbol{r}_2^*$. Therefore, according to Definition 2.4, $\mathrm{y}^* \mid \{\mathbf{x}, \mathbf{r}^*\}$ is strictly monotonic with respect to $\mathbf{r}^*$.

If $\mathrm{y}^* \mid \{\mathbf{x}, \mathbf{r}^*\}$ is monotonic with respect to $\mathbf{r}^* = [-\mathrm{t}, \mathbf{r}]$, then for any vectors $\boldsymbol{r}_1 \prec \boldsymbol{r}_2$ and $s \in \mathrm{Dom}(\mathrm{y})$, the inequality $\Pr(\mathrm{y}^* = 1 | \mathbf{x}, \mathbf{r}^* = [-s, \boldsymbol{r}_1]) < \Pr(\mathrm{y}^* = 1 | \mathbf{x}, \mathbf{r}^* = [-s, \boldsymbol{r}_2])$ holds, thereby ensuring that $\Pr(\mathrm{y} > s | \mathbf{x}, \mathbf{r} = \boldsymbol{r}_1) < \Pr(\mathrm{y} > s | \mathbf{x}, \mathbf{r} = \boldsymbol{r}_2)$, which confirms the monotonicity of y with respect to $\mathbf{r}$. $\square$

## B.2. Proof of Lemma 2

*Proof.* If $\Pr(\mathrm{y} = 1 | \mathbf{x}, \mathbf{r})$ is monotonically increasing with respect to $\mathbf{r}$. We define $G(\mathbf{x}, \mathbf{r}) = \Pr(\mathrm{y} = 1 | \mathbf{x}, \mathbf{r})$, then the function $G_{\boldsymbol{x}_0}(\mathbf{r}) = G(\mathbf{x} = \boldsymbol{x}_0, \mathbf{r}) \in (0, 1)$ is continuous and monotonically increasing with respect to $\mathbf{r}$ within the bounded set $D$. Therefore, $G_{\boldsymbol{x}_0}(\mathbf{r})$ is a fraction (by $D$) of a cumulative distribution function, and we still use $G_{\boldsymbol{x}_0}$ to denote this CDF. Define the random variable c such that $(\mathbf{c} \mid \mathbf{x} = \boldsymbol{x}_0) \sim G_{\boldsymbol{x}_0}(\mathbf{c})$, which is conditionally independent of the variable $\mathbf{r}$, i.e.

$\mathbf{r} \perp\!\!\!\perp \mathbf{c} \mid \mathbf{x}$. Furthermore, we find that

$$
\begin{aligned}
&\Pr(\mathbf{c} \prec \mathbf{r}|\mathbf{x} = \boldsymbol{x}_0, \mathbf{r} = \boldsymbol{r}_0) \\
&= \Pr(\mathbf{c} \prec \boldsymbol{r}_0|\mathbf{x} = \boldsymbol{x}_0, \mathbf{r} = \boldsymbol{r}_0) \\
&= \Pr(\mathbf{c} \prec \boldsymbol{r}_0|\mathbf{x} = \boldsymbol{x}_0) \qquad \text{(by } \mathbf{r} \perp\!\!\!\perp \mathbf{c} \mid \mathbf{x}) \\
&= G_{\boldsymbol{x}_0}(\mathbf{r}_0) \\
&= G(\boldsymbol{x}_0, \boldsymbol{r}_0) \\
&= \Pr(\mathbf{y} = 1|\mathbf{x} = \boldsymbol{x}_0, \mathbf{r} = \boldsymbol{r}_0)
\end{aligned}
$$

thus $\mathbf{y} \mid \{\mathbf{x}, \mathbf{r}\} \overset{d}{=} \mathbb{I}(\mathbf{c} \prec \mathbf{r}) \mid \{\mathbf{x}, \mathbf{r}\}$.

Conversely, suppose that there exists a variable $\mathbf{c}$ such that $\mathbf{r} \perp\!\!\!\perp \mathbf{c} \mid \mathbf{x}$ and $\mathbf{y} \mid \{\mathbf{x}, \mathbf{r}\} \overset{d}{=} \mathbb{I}(\mathbf{c} \prec \mathbf{r}) \mid \{\mathbf{x}, \mathbf{r}\}$. For any $\boldsymbol{r}_1 \prec \boldsymbol{r}_2$ where $\boldsymbol{r}_1, \boldsymbol{r}_2 \in D$, it follows:

$$
\begin{aligned}
&\Pr(\mathbf{y} = 1|\mathbf{x}, \mathbf{r} = \boldsymbol{r}_1) \\
&= \Pr(\mathbf{c} \prec \mathbf{r}|\mathbf{x}, \mathbf{r} = \boldsymbol{r}_1) \quad \text{(by } \mathbf{y} \mid \{\mathbf{x}, \mathbf{r}\} \overset{d}{=} \mathbb{I}(\mathbf{c} \prec \mathbf{r}) \mid \{\mathbf{x}, \mathbf{r}\}) \\
&= \Pr(\mathbf{c} \prec \boldsymbol{r}_1|\mathbf{x}) \qquad\qquad\qquad\qquad \text{(by } \mathbf{r} \perp\!\!\!\perp \mathbf{c} \mid \mathbf{x}) \\
&\prec \Pr(\mathbf{c} \prec \boldsymbol{r}_2|\mathbf{x}) \qquad\qquad\qquad\qquad \text{(by } \boldsymbol{r}_1 \prec \boldsymbol{r}_2) \\
&= \Pr(\mathbf{y} = 1|\mathbf{x}, \mathbf{r} = \boldsymbol{r}_2),
\end{aligned}
$$

confirming that $\Pr(\mathbf{y} = 1|\mathbf{x}, \mathbf{r})$ is monotonically increasing with respect to $\mathbf{r}$. $\qquad\square$

## C. Discussion on Bounded Revenue Variable

The bounded revenue variable issue can arise in practice, for instance, when our model is trained with $\|\mathbf{r}\| < b$, but is used for inference with revenue $\mathbf{r}$ whose elements $\mathbf{r}_i \gg b$. This discrepancy can cause an out-of-distribution (OOD) problem, resulting in $\Pr(\mathbf{y} = 1|\mathbf{x}, \mathbf{r}) = \Pr(\mathbf{r} \succ \mathbf{c}|\mathbf{x}) \to 1$. We address this by hypothesizing a non-zero probability $\Pr(\mathbf{c} = +\infty|\mathbf{x}) = p_0 > 0$, which yields a bounded estimate:

$$
\Pr(\mathbf{y} = 1|\mathbf{x}, \mathbf{r}) = \Pr(\mathbf{r} \succ \mathbf{c}|\mathbf{x}) < 1 - p_0.
$$

This upper bound is independent of $\mathbf{r}$. Alternatively, we can replace the revenue variable $\mathbf{r}$ with a bounded invertible monotonic function $h(\mathbf{r})$, for instance, the sigmoid function. Consequently, the probability $\Pr(\mathbf{y} = 1|\mathbf{x}, h(\mathbf{r}))$ will be monotonic with respect to $h(\mathbf{r})$, effectively transforming the original unbounded problem into a bounded one.

## D. Details of GCM

### D.1. Derivation of ELBO

The ELBO of GCM is expressed as follows:

$$
\begin{aligned}
&\log p_\theta(\mathbf{x}, \mathbf{r}, \mathbf{y}) \\
&= \log \mathbb{E}_{\mathbf{z} \sim p(\mathbf{z})} p_\theta(\mathbf{x}, \mathbf{r}, \mathbf{y}|\mathbf{z}) \\
&= \log \mathbb{E}_{\mathbf{z} \sim q_\phi} \frac{p_\theta(\mathbf{x}, \mathbf{r}, \mathbf{y}|\mathbf{z}) p_\theta(\mathbf{z})}{q_\phi(\mathbf{z}|\mathbf{x})} \\
&\geq \mathbb{E}_{\mathbf{z} \sim q_\phi} \log \frac{p_\theta(\mathbf{x}, \mathbf{r}, \mathbf{y}|\mathbf{z}) p_\theta(\mathbf{z})}{q_\phi(\mathbf{z}|\mathbf{x})} \\
&= \mathbb{E}_{\mathbf{z} \sim q_\phi} \log \frac{\Pr_\theta(\mathbf{c} \curlyvee_\mathbf{y} \mathbf{r}|\mathbf{z}, \mathbf{r}) p_\theta(\mathbf{x}, \mathbf{r}|\mathbf{z}) p_\theta(\mathbf{z})}{q_\phi(\mathbf{z}|\mathbf{x})} \\
&= \mathbb{E}_{\mathbf{z} \sim q_\phi} \log \frac{\Pr_\theta(\mathbf{c} \curlyvee_\mathbf{y} \mathbf{r}|\mathbf{z}) p_\theta(\mathbf{z})}{q_\phi(\mathbf{z}|\mathbf{x})} + \log p_\theta(\mathbf{x}, \mathbf{r}|\mathbf{z}).
\end{aligned}
$$

## D.2. Calculate the Likelihood

We assume that $\mathbf{c} \mid \mathbf{z}$ is element-wise independent. Thus, we can decompose the conditional likelihood as follows:

$$\Pr{}_\theta(\mathbf{c} \curlyvee_\mathsf{y} \mathbf{r}|\mathbf{z}, \mathbf{r}) = 1 - \mathsf{y} - (-1)^\mathsf{y} \prod_i \int_{-\infty}^{\mathbf{r}_i} p_\theta(\mathbf{c}_i|\mathbf{z})d\mathbf{c}_i, \tag{10}$$

where $\mathbf{c}_i$ is the $i$-th element of $\mathbf{c}$ and $\mathbf{r}_i$ is the $i$-th element of $\mathbf{r}$.

## D.3. GCM for Continuous Regression

When y is a continuous variable, the regression problem can be transformed into a binary classification problem, as demonstrated in Lemma 4.1. Consequently, this allows us to develop the generative model for the latent variables t and $\mathbf{c}$, so that

$$\Pr(\mathsf{y} > \mathsf{t}|\mathbf{x}, \mathbf{r}, \mathbf{z}) = \Pr(\mathbf{r} \succ \mathbf{c}|\mathbf{x}, \mathbf{r}, \mathbf{z}) = \Pr(\mathbf{r} \succ \mathbf{c}|\mathbf{r}, \mathbf{z}). \tag{11}$$

We assume that y and t adhere to conditional Gaussian distributions: specifically, $p_\theta(\mathsf{y}|\mathbf{x}, \mathbf{r}, \mathbf{z}) = \mathcal{N}\left(\mathsf{y}; \mu_\mathsf{y}(\mathbf{x}, \mathbf{r}, \mathbf{z}), \sigma_\mathsf{y}^2(\mathbf{x})\right)$ and $p_\theta(\mathsf{t}|\mathbf{x}, \mathbf{r}, \mathbf{z}) = \mathcal{N}\left(\mathsf{t}; \mu_\mathsf{t}(\mathbf{x}), \sigma_\mathsf{t}^2(\mathbf{x})\right)$. Here, $\sigma_\mathsf{y}(\mathbf{x})$, $\mu_\mathsf{t}(\mathbf{x})$, and $\sigma_\mathsf{t}(\mathbf{x})$ are functions we defined, while $\mu_\mathsf{y}(\mathbf{x}, \mathbf{r}, \mathbf{z})$ is determined via Equation (11). We recognize that

$$\Phi\left(\frac{\mu_\mathsf{y}(\mathbf{x}, \mathbf{r}, \mathbf{z}) - \mu_\mathsf{t}(\mathbf{x})}{\sqrt{\sigma_\mathsf{y}^2(\mathbf{x}) + \sigma_\mathsf{t}^2(\mathbf{x})}}\right) = \Pr(\mathsf{y} > \mathsf{t}|\mathbf{x}, \mathbf{r}, \mathbf{z}) = \Pr(\mathbf{r} \succ \mathbf{c}|\mathbf{r}, \mathbf{z}),$$

enabling us to define $\mu_\mathsf{y}$ as

$$\mu_\mathsf{y}(\mathbf{x}, \mathbf{r}, \mathbf{z}) = \sqrt{\sigma_\mathsf{y}^2(\mathbf{x}) + \sigma_\mathsf{t}^2(\mathbf{x})}\left\{\Phi^{-1}\left(\Pr(\mathbf{r} \succ \mathbf{c}|\mathbf{r}, \mathbf{z})\right)\right\} + \mu_\mathsf{t}(\mathbf{x}).$$

The established definitions of y and t ensure that t is conditionally independent of y given $\mathbf{x}$ and $\mathbf{r}$, as stipulated by Lemma 4.1. By employing variational inference, the loss function corresponding to the continuous problem is expressed as:

$$\mathcal{L} = -\mathbb{E}_{\mathbf{z} \sim q_\phi} \log \frac{p_\theta(\mathsf{y}|\mathbf{z}, \mathbf{r})p_\theta(\mathbf{z})\,p_\theta(\mathbf{x}|\mathbf{z})}{q_\phi(\mathbf{z}|\mathbf{x})}$$

$$= \mathbb{E}_{\mathbf{z} \sim q_\phi}\left\{\frac{(\mathsf{y} - \mu_\mathsf{y})^2}{2\sigma_\mathsf{y}{}^2} + \log \sigma_\mathsf{y} - \log \frac{p_\theta(\mathbf{z})\,p_\theta(\mathbf{x}|\mathbf{z})}{q_\phi(\mathbf{z}|\mathbf{x})}\right\}.$$

The estimator for y with respect to $\mathbf{x}$ and $\mathbf{r}$ is expressed as:

$$\hat{y} = \frac{1}{N}\sum_{n=1}^{N} \mu_\mathsf{y}(\mathbf{x}, \mathbf{r}, \mathbf{z} = \mathbf{z}^{(n)}),$$

where $\mathbf{z}^{(n)}$ is sampled randomly from the variational distribution $q_\phi(\mathbf{z}|\mathbf{x})$.

# E. Experiment Details

## E.1. Positive Weight Matrix

In both the baseline models and our IGCM approach, positive matrices are essential to ensure monotonic linear projections. Through our experiments, we discovered that the transformation:

$$\mathbf{W}_+ = \frac{\mathrm{softplus}(10\mathbf{W})}{10},$$

yields better positive matrices compared to using operators like square, softplus, leaky relu, or exp.

## E.2. Hyperparameters

Table 5. Hyperparameters of the experiments.

| dataset | hidden dim. | sample number | latent dim. | max epcoh | optimizer | batch size | learning rate |
|---|---|---|---|---|---|---|---|
| Adult | 16 | 32 | 4 | 40 | ADAM(Kingma & Ba, 2015) | 256 | 0.001 |
| COMPAS | 6 | 32 | 6 | 200 | ADAM | 64 | 0.01 |
| Diabetes | 16 | 32 | 4 | 20 | SGD | 256 | 0.3 |
| Blog Feedback | 16 | 32 | 4 | 40 | ADAM | 256 | 0.001 |
| Loan Defaulter | 16 | 32 | 4 | 20 | ADAM | 512 | 0.003 |
| Auto MPG | 6 | 32 | 3 | 120 | ADAM | 16 | 0.01 |

# F. Ablation Studies

## F.1. Ablation on Latent Dimension and Sample Number

We conduct ablation studies on the GCM and IGCM methods using the Adult dataset, focusing on two key hyperparameters: $D$, representing the latent dimensionality, and $N$, the sampling amount. The values for $D$ and $N$ are chosen from the sets $\{4, 8, 12, 16\}$ and $\{8, 16, 24, 32\}$ respectively, and each experiment is repeated 10 times with distinct random seeds. The results indicate that both GCM and IGCM perform optimally in the parameter space's northeast region.

Table 6. Experimental results (AUC) of GCM on the Adult dataset with multiple $D$ and $N$ settings.

|  | $D = 4$ | $D = 8$ | $D = 12$ | $D = 16$ |
|---|---|---|---|---|
| $N = 8$ | 0.7834 ±0.0026 | 0.7840 ±0.0028 | 0.7842 ±0.0025 | 0.7857 ±0.0022 |
| $N = 16$ | 0.7833 ±0.0024 | 0.7824 ±0.0023 | 0.7835 ±0.0015 | 0.7851 ±0.0024 |
| $N = 24$ | 0.7833 ±0.0014 | 0.7825 ±0.0021 | 0.7816 ±0.0022 | 0.7842 ±0.0021 |
| $N = 32$ | 0.7844 ±0.0025 | 0.7822 ±0.0030 | 0.7836 ±0.0024 | 0.7843 ±0.0034 |

Table 7. Experimental results (AUC) of IGCM on the Adult dataset with multiple $D$ and $N$ settings.

|  | $D = 4$ | $D = 8$ | $D = 12$ | $D = 16$ |
|---|---|---|---|---|
| $N = 8$ | 0.7892 ±0.0019 | 0.7900 ±0.0022 | 0.7907 ±0.0015 | 0.7914 ±0.0010 |
| $N = 16$ | 0.7894 ±0.0024 | 0.7911 ±0.0012 | 0.7909 ±0.0009 | 0.7918 ±0.0009 |
| $N = 24$ | 0.7908 ±0.0018 | 0.7901 ±0.0020 | 0.7924 ±0.0007 | 0.7893 ±0.0019 |
| $N = 32$ | 0.7891 ±0.0018 | 0.7899 ±0.0018 | 0.7889 ±0.0022 | 0.7917 ±0.0011 |

# G. Comparison of Time Complexity

One of the primary benefits of our GCM model is its efficiency during the inference phase. For a given $\mathbf{x}$, the model efficiently computes $p_\theta(\mathrm{y}|\mathbf{x} = \boldsymbol{x}, \mathbf{r} = \boldsymbol{r}_i)$ across various $\mathbf{r}$ values. This efficiency stems from the fact that the GCM model predicts the latent variables $\mathbf{z}$ and $\mathbf{c}$ based solely on $\mathbf{x}$. This enables it to infer y using $\mathbf{c}$ and $\mathbf{r}$ as per Equation (10) with minimal computational time. Consequently, we bypass the extensive time required to process each pair $(\mathbf{x} = \boldsymbol{x}, \mathbf{r} = \boldsymbol{r}_i)$ through a deep neural network, as is common with traditional approaches. We assessed the inference efficiency for different counts of $\mathbf{r}$ while maintaining a constant $\mathbf{x}$, with results detailed in Table 8. As illustrated, the GCM emerges as the fastest method when the quantity of $\mathbf{r}$ surpasses 64, highlighting its efficiency in multi-revenue prediction contexts. When the count of $\mathbf{r}$ reaches a peak value of $1,024$, GCM can reduce computational time by up to $72\%$ compared to the quickest baseline model.

*Table 8.* Inference time cost (ms per batch) of different models with different numbers of **r** on the COMPAS dataset.

| Method | Number of r | | | | | | | | | | |
|---|---|---|---|---|---|---|---|---|---|---|---|
| | 1 | 2 | 4 | 8 | 16 | 32 | 64 | 128 | 256 | 512 | 1024 |
| MM | 1.51 | 2.35 | 3.33 | 4.83 | 9.27 | 17.36 | 31.24 | 58.53 | 112.65 | 306.57 | 308.33 |
| CMNN | 3.39 | 5.17 | 9.02 | 15.87 | 28.95 | 51.96 | 102.01 | 198.07 | 394.63 | 869.76 | 877.47 |
| PWL | **1.02** | **1.67** | **2.47** | **3.73** | **7.86** | **13.89** | 26.01 | 47.86 | 92.95 | 280.70 | 285.48 |
| GCM | 11.66 | 11.55 | 11.98 | 12.89 | 13.88 | 16.85 | **20.14** | **28.89** | **43.88** | **76.23** | **79.63** |

