# OpenReview forum: "Learning Monotonic Probabilities with a Generative Cost Model"
_ICML.cc/2025/Conference — ICML 2025 poster_

### Official Review · Reviewer_6X1f · 2025-03-11

**Overall Recommendation:** 3

**Summary:**

The paper tackles the problem of enforcing monotonicity in predictions as a function of some variables if the true underlying function also abides by the monotonicity argument. Given the prediction target $y$ that is supposed to be monotonic with respect to the variable $r$, the paper relies on intuitive observation that this monotonicity condition can be translated to some additional auxiliary variable as long as that variable satisfies conditional independence condition. The paper then proposes a generative model to learn this auxiliary variable using variational inference. Experiments are done on synthetic dataset as well other real-world datasets.

**Claims And Evidence:**

1. Theoretical claims are well supported, and the paper is generally well-motivated in that sense.
2. I like the synthetic experiment on quantile regression to tractably study the problem setting, however I could not verify the insights in Figure 6. The paper claims estimated quantile values maintain strict monotonicity; I think I'd be helped if authors can provide an elaborate and annotated version of Figure 6 so that I can compare the violations of monotonicity for baselines methods.
3. In a similar sense, I think I'd be helped if more intuition can be provided for the real-world dataset experiments in Section 5.2. The paper currently compares standard metrics like AUC and RMSE, but I'd appreciate if authors could clarify how the monotonicity assumption or requirement factors in that improves these metrics as the paper claims or the experiments suggest.

**Essential References Not Discussed:**

Not that I'm familiar of.

**Experimental Designs Or Analyses:**

Some questions remain (see above).

**Methods And Evaluation Criteria:**

I like the synthetic experiment to tractably study, and it makes a lot of sene to study quantile regression as a test-bed for monotonicity (however, I believe the experimental insights can be presented more meaningfully), and would appreciate clarification on the evaluation on real-world datasets (see above).

I'd personally also like to see comparisons on the efficiency or the computational aspects of the inference mechanism as proposed in the paper compared to the baselines. I see Table 2, and to me, the reported numbers live close to each other, or the improvements are not that significant; I don't want to play the devil here as I don't necessarily think that's a strong concern, but I'd love to get some more insights as what are other potential improvements of the proposed methodology; computational, ease of applicability, etc.

**Other Comments Or Suggestions:**

None.

**Other Strengths And Weaknesses:**

The paper is generally written well, but maybe proofs can go to appendix, to make space for more experimental insights. In addition to that, the paper is bit heavy on the notation, and could use some intuition to motivate the main insights. And maybe positioning it more broadly it into the current literature could help strengthen the contributions.

**Questions For Authors:**

See above.

**Relation To Broader Scientific Literature:**

The paper presents an intuitive insight and builds an inference mechanism around it to enforce monotonicity in the probabilistic predictions. Compared to the literature in the paper, it

**Theoretical Claims:**

While I do get the intuition behind their theoretical claims (Lemma 4.1 and Lemma 4.2), I haven't very rigorously verified the notational description of the proofs. But to me the results are basic (in a positive sense) and correct.

---

> ### Author Rebuttal · Authors · 2025-03-31
>
> Thank you for the reviews and valuable suggestions. Here are our responses to the concerns raised:
>
> # Answer 1: Clarification on Figure 6
>
> The scatter plot in the background represents the training instances $(x_i, y_i)$, with $p(y|x)$ defined in the southeast corner of page 7. Consider $y_r|x$ as the r-th quantile of $y$ given $x$; thus, $y_r$ is a function of $x$, and $(x, y_r)$ can form a curve $\\Gamma_r$ on the x-y plane. For $r_2>r_1$, the monotonicity of quantiles ensures that the curve $\\Gamma_{r_2}$ lies above the curve $\\Gamma_{r_1}$. In Figure 6, five estimated quantile curves (red curves) are plotted for $r$ in $(0.1, 0.3, 0.5, 0.7, 0.9)$ using different monotonic modeling methods at different training stages. Each column signifies a distinct modeling approach, and each row denotes a training stage. An accurate quantile value approximation should result in each estimated quantile curve being close to its actual quantile curve. Therefore, the estimated curves should:
> - Not have excessively narrow gaps between them.
> - Have $\\Gamma_{0.1}$ and $\\Gamma_{0.9}$ adjacent to the edges of the background scatter plot, with limited outliers.
>
> Observing Figure 6 reveals that:
> - The red curves for PosNN, MM, SMM, and SMNN (mostly monotonic by construction methods) are clustered in certain areas.
> - CMNN, Hint, and PWL (mostly monotonic by regularization methods) exhibit too many outliers.
> - GCM displays distinct gaps between the red curves and the fewest outliers.
>
> # Answer 2: More Intuition for the Experiments
>
> **Comparison between GCM and IGCM**: Our analysis reveals that within public datasets, the IGCM outperforms the GCM. This improved performance is attributed to the fact that real-world scenarios do not always adhere to strict monotonic relationships; for instance, while BMI is correlated with diabetes, it is not necessarily causative. Typically, the selection of monotonic features is informed by experience or statistical analysis. Under these circumstances, implicit monotonic assumptions tend to be more applicable. Our experimental data bolsters this assertion, as IGCM demonstrates superior performance to GCM in five of the six datasets.
>
> **Comparison between strict monotonic methods**: The strict monotonic methods evaluated include PosNN, MM, SMM, CMNN, SMNN, and the proposed GCM. The basic PosNN shows poor performance across most datasets, indicating that merely establishing a positively weighted neural network is insufficient for monotonic problems. Interestingly, the traditional MM method performs commendably, and its successor, SMM, exhibits a correlation, indicating that the core min-max architecture proves effective in optimizing the monotonic network. Recent developments like CMNN and SMNN introduce innovative monotonic network architectures, outperforming MM in several experiments. Our proposed GCM method consistently exhibits strong performance across all datasets, suggesting that reframing the monotonic problem as a generative one yields enhanced results compared to simply altering the network structure.
>
> **Comparison between nonstrict monotonic methods**: Hint, PWL, and IGCM are non-strict monotonic methods, yet only IGCM addresses the non-strict issue through implicit monotonic modeling. This approach consists of two strict monotonic challenges: $p(r|k)$ and $p(y|k)$, establishing a nonstrict monotonic link between $y$ and $r$, which proves beneficial when compared to traditional non-strict monotonic methods.
>
> # Answer 3: Other Potential Improvements
>
> **Rapid Decision Making**: As illustrated in Appendix D, GCM and IGCM provide the quickest inference for multiple revenue variables, providing a clear advantage in decision-making models. For instance, consider a robot tasked with determining a sequence of actions for a particular mission, where each action is directly proportional to total energy usage. If there is a model capable of forecasting energy consumption to help the robot minimize energy loss, this model must rapidly predict multiple action options. Here, the GCM method proves beneficial for swift decision making.
>
> **Scalability**: GCM is inherently scalable. Unlike previous monotonic models that struggle to scale due to restrictions on non-monotonic normalization techniques (like LN, BN) and limitations on activation functions or linear projections, GCM can incorporate any structure to model $p(c|x,z)$ and $q(z|x)$. Thus, with more extensive datasets, GCM's scalability advantages may become more evident.
>
> **Monotone Multitask Modeling**: For a set of tasks $y_1,\\cdots,y_n$ that are monotonic such that $(y_i=1)\\subset(y_{i+1}=1)$ and each $y_i$ is monotonic with respect to $r$, we can apply the monotonic cost variable sequence $c_1\\succ \\cdots \\succ c_k$ and let $y_i=\\mathbb I(r \succ c_i)$ to address it.
>
> We hope these clarifications address your concerns and demonstrate the validity and applicability of our proposed method. Thank you again for your valuable feedback.

---

> > ### Comment · Reviewer_6X1f · 2025-04-04
> >
> > thanks for the response. I'm happy to keep my score.

---

### Official Review · Reviewer_fPoj · 2025-03-12

**Overall Recommendation:** 1

**Summary:**

The paper studies the problem of monotonic regression where the target variable should maintain a monotonous relationship with part of the input variables. It first establishes some analytical properties for the probability model underlying the monotonous regression and then it proposes a Bayesian network model to capture the relationships between different classes of variables. It provides numerical experiments to support the discussion.

**Claims And Evidence:**

Yes. I think all the claims are supported.

**Essential References Not Discussed:**

Nan.

**Experimental Designs Or Analyses:**

The paper didn't discuss much about the choice of functions in (4) and (8).

It mentions "All methods utilize the same foundational architecture: a three-layer perceptron network utilizing tanh activations". I think this part is done too casually. Is a three-layer network sufficient to serve as a generative model?

**Methods And Evaluation Criteria:**

The evaluation criteria make sense to me.

But I disagree with the proposed method. If we step back, the monotonous regression can be viewed as a discriminative task, though there is the challenge of maintaining monotonicity. Yet the proposed solution essentially relies on a generative model, or, a probability model for the joint distribution of input variables and output variables, together with some latent variables. Generally, it's perceived that generative modeling tasks are more difficult than their discriminative counterparts. So I think the proposed method is to some extent misleading for future works or practical uses, in that it is too ambitious to assume that the joint probability distribution can be approximated/learned by the proposed probability model, and there is no way to know/check when it cannot be.

**Other Comments Or Suggestions:**

See above.

**Other Strengths And Weaknesses:**

Nan.

**Questions For Authors:**

See above.

**Relation To Broader Scientific Literature:**

Nan.

**Theoretical Claims:**

Yes, the claims are correct based on my check.

---

> ### Author Rebuttal · Authors · 2025-03-31
>
> Thank you for your detailed feedback and suggestions. We appreciate the opportunity to address the concerns raised in the review.
>
> # Answer 1: Discriminative vs. Generative Models
>
> We respectfully disagree with the assertion that generative models are inherently more challenging than discriminative models. The modeling difficulty depends on the task and the structure used. For example, naive Bayes, a generative model, is widely used for classification tasks and is comparable to discriminative methods such as logistic regression. In our paper, we have shown that a discriminative model $\\Pr(y=1|x,r)=G(x,r)$ can be transformed into a generative model $p(x,c)=p(c|x)p(x)$ using the relationship established in Lemma-2:
> $$
> \\Pr( c \\prec  r| x) =\\Pr( c \\prec  r| x,r) = G(x,  r),
> $$
> and consequently
> $$
> p( c | x) =\\partial G(x,  c) / \\partial c.
> $$
> This demonstrates that in monotonic modeling tasks, these approaches can be interconverted, so it is unfair to say that generative methods in monotonic modeling are more difficult arbitrarily.
>
> Moreover, traditional monotonic networks face challenges such as requiring all weight matrices to be positive [1] and all activations to be monotonic and not fully convex or concave [2]. Common normalization techniques like layer-norm and batch-norm are also not allowed since they are not monotonic operators. As a result, it is hard to train a deep monotonic network. However, with the cost generative model, we can easily build a deep network for generating the cost variable $c$ and obtain the target variable $y$ via $\\{y=1\\}=\\{c\prec r\\}$. This allows us to use unconstrained weight matrices, non-monotonic activations such as silu and gelu, fully convex activations such as relu and softplus, and normalization techniques widely used in deep networks such as LN and BN. Training this generative network is also straightforward following the loss function demonstrated in (7).
>
> [1] Monotonic Networks. [Link](https://proceedings.neurips.cc/paper_files/paper/1997/file/83adc9225e4deb67d7ce42d58fe5157c-Paper.pdf)
>
> [2] Constrained Monotonic Neural Networks. [Link](https://arxiv.org/pdf/2205.11775)
>
> # Answer 2: Challenge of Learning the Joint Probability Distribution
>
> Previous studies show the outstanding performance of learning the joint probability in machine learning tasks, such as learning the joint distribution of an image via generative methods. For simple tasks such as the MNIST dataset, a simple four-layer VAE [3] is sufficient, while more complex tasks like CIFAR10 might require deeper models such as DDPM [4]. This paper has shown that the cost variable $c$ is essential for all monotonic problems $\\Pr(y=1|x, r)$, where $r$ is the monotonic revenue variable, $x$ is the non-monotonic variable, and $y$ is the response variable determined by $c\\prec r$. Since $x,r,c$ are all real vectors, we can adopt model structures similar to image generative models.
>
> To design the generative model for $x,r,c$, we have chosen a basic structure similar to the VAE, as the tasks in our paper are not overly complex. Although $c$ is not directly observable, we can evaluate the model for $c$ using the observable variable $y=\\mathbb I(r\\succ c)$, and the testing AUC/ACC of $y$ indicates the performance of our generative model of $c$. Additionally, the ELBO $\\mathbb E_{z\\sim q}\\log p(x,r,y|z)-D_{KL}(q(z|x)\\|p(z))\\leq \\log p(x,r,y)$ serves as a lower bound for the evidence $\\log p(x,r,y)$, providing another evaluation metric.
>
> [3] Auto-Encoding Variational Bayes. [Link](https://arxiv.org/abs/1312.6114)
>
> [4] Denoising Diffusion Probabilistic Models. [Link](https://arxiv.org/abs/2006.11239)
>
> # Answer 3: Choice of Functions in (4) and (8)
>
> We employ the standard normal distribution for the priors of $z_1$ and $z_2$. Each conditional likelihood $p(a|b)$ is expressed as:
> $$
> \\mu_a, \\log \\sigma_a = \\text{MLP}(b), \\ \\
> a \\sim \\mathcal N (\\mu_a, \\sigma_a^2).
> $$
> In the GCM model, we use the reparameterization trick to derive the variable $a$, or we apply the CDF function of the normal distribution to compute the probability $\\Pr(a<a_0)$. The MLP here consists of one or two layers. We will provide further details in the appendix of future revisions.
>
> # Answer 4: Sufficiency of a Three-Layer Network
>
> The number of layers is not a constraint for generative models. As mentioned in Answer 2, the VAE applied to the MNIST dataset uses only two layers with tanh activation functions and two affine layers without activations, yet it successfully generates high-quality handwritten numerals. In our experiments, the dimension of the revenue variable (see Table 3 on page 8) is less than 10, much smaller than the pixel count of an MNIST image. Therefore, we chose not to use an excessively deep generative network for our study.
>
> We hope these clarifications address your concerns and demonstrate the validity and applicability of our proposed method. Thank you again for your valuable feedback.

---

### Official Review · Reviewer_rxdb · 2025-03-13

**Overall Recommendation:** 4

**Summary:**

The paper introduces a new generative framework to model monotonic probabilities by reformulating the traditional problem into learning a latent cost variable. Instead of directly designing a monotonic function, the authors propose that for a binary outcome, the probability is given by the event that a latent cost variable is dominated by a revenue variable. Two models are presented: (1) GCM (Generative Cost Model): Targets strict monotonicity by modeling the latent cost variable via two independent latent variables. (2) IGCM (Implicit Generative Cost Model): Extends the approach to capture cases where monotonicity reflects correlation rather than a strict order.

The paper validates the approach through simulated quantile regression and experiments on multiple public datasets. Results indicate that GCM/IGCM achieve superior performance compared to existing monotonic neural network methods.

**Claims And Evidence:**

I think authors argument are clear and convincing.

**Essential References Not Discussed:**

Although I am not familiar with the relevant literature, the references cited in the paper provide sufficient context to help me understand its contributions.

**Experimental Designs Or Analyses:**

I think the experimental design and analyses are sound and effectively demonstrate the strengths of the proposed models.

**Methods And Evaluation Criteria:**

I think both the proposed methods and the evaluation criteria are sensible for addressing monotonicity in probabilistic predictions.

**Other Comments Or Suggestions:**

N/A

**Other Strengths And Weaknesses:**

- Strengths : The paper’s main strengths lie in its originality and rigor. It introduces a novel formulation by recasting the monotonic probability problem as one of learning a latent cost variable, thereby addressing both strict and implicit monotonicity within a unified generative framework. Their contributions supported by solid theoretical derivations and extensive experiments on simulated and real-world datasets.

- Weaknesses:  Assumptions such as a bounded revenue variable and conditional independence, may not always align with practical scenarios. I am wondering how sensitive of the methods to these assumptions, what would happen if the model assumptions are misspecified?

**Questions For Authors:**

1. Could the authors clarify the practical implications of the conditional independence assumption? In particular, what types of real-world datasets or scenarios might violate this condition, and how does the model’s performance degrade in such cases?

2. The model involves variational inference with multiple latent variables, and the results appear sensitive to hyperparameter choices. Could the authors provide additional ablation studies on how these choices affect both training scalability and performance of the method?

**Relation To Broader Scientific Literature:**

I think the authors successfully relate their contributions to prior findings and articulate how their work extends and improves upon existing methods.

**Theoretical Claims:**

The proofs are logically coherent, I think they make sense to me.

---

> ### Author Rebuttal · Authors · 2025-03-31
>
> Thank you for the reviews and valuable suggestions. Here are our answers to the concerns raised:
>
> # Answer 1: Bounded Revenue Variable
>
> This issue can arise in practice, for instance, when our model is trained with $r \\in (-10, 10)$, but is used for inference with values $r \\gg 10$. Such a discrepancy can lead to an out-of-distribution (OOD) problem, resulting in $\\Pr(y=1|x,r)=\\Pr(r>c|x)\\to1$. We address this by hypothesizing a non-zero probability $\\Pr(c=+\\infty|x)=p_0>0$, which yields a bounded estimate:
> $$
> \\Pr(y=1|x,r)=\\Pr(r>c|x)<1-p_0.
> $$
> This upper bound is independent of $r$. Alternatively, we can replace the revenue variable $r$ with a bounded monotonic function $h(r)$. Consequently, the probability $\\Pr(y=1|x,h(r))$ will be monotonic with respect to $h(r)$, effectively transforming the original unbounded problem into a bounded one.
>
> # Answer 2: Conditional Independence
>
> The conditional independence $c\\perp r\\mid x$ is crucial for solving the strict monotonic issue of $p(y|x,r)$ by achieving $\\Pr(c\\prec r|x,r)=\\Pr(c\\prec r|x)$, with $r$ representing the revenue variable and $y$ as its monotonic outcome. As demonstrated in Lemma-2, for any strictly monotonic probability $p(y|x,r)$, such a latent variable $c$ is guaranteed to exist, allowing us to construct the $c$ model without worrying about its practical interpretation.
>
> Conversely, in non-strict monotonic problems, finding such a $c$ is impossible. For example, a consumer might prefer buying a product at a lower price, but if the price is excessively low, it might seem dubious, resulting in the consumer not buying it, illustrating that $c$ (consumer cost) is affected by $r$ (inverse of the product price). In this case, an implicit monotonic model, detailed in section-4.4, can be utilized. Here, the concealed kernel variable $k$ denotes the product's authenticity, and then $r$ shows monotonicity with respect to $k$, enabling us to identify a cost variable $c$ so that $c\\perp k\\mid x$, instead of $c\\perp r\\mid x$. Here, $c$ and $k$ are both latent variables that the modeling for them will not obey the practical interpretation of $r$ and $y$.
>
> # Answer 3: Ablation on Hyperparameters in Variational Inference
>
> The ablation of $D$, the latent variable dimension, and $N$, the sampling number, is detailed in Table-5 and Table-6 within Appendix C. Testing results for GCM using the Adult dataset are as follows:
> |      | D=4 | D=8 | D=12 | D=16 |
> | ---- | ------ | ---- | ----- | ---- |
> | N=8  | 0.7834 | 0.7840 | 0.7842 | 0.7857 |
> | N=16 | 0.7833 | 0.7824 | 0.7835 | 0.7851 |
> | N=24 | 0.7833 | 0.7825 | 0.7816 | 0.7842 |
> | N=32 | 0.7844 | 0.7822 | 0.7836 | 0.7843 |
>
> The findings indicate that $D$, the latent dimension, impacts the test AUC, whereas the sampling number $N$ plays a lesser role. Another critical hyperparameter is $\\beta$, utilized in the ELBO as per the $\\beta$VAE approach, where the modified ELBO is expressed as follows,
> $$
> \\text{ELBO}_\\beta=\\mathbb E_q \\log {\\text{Pr}( c\\curlyvee_y r| z, r)p({x},{r}| z)}-\\beta \\log \\mathbb E_q \\frac{q( z| x)}{p( z)}.
> $$
> The test outcomes for varying values of $\\beta$ are:
> |model |AUC |
> |----|-----|
> |$\\beta=0$ |0.7844|
> |$\\beta=1$ |0.7828|
> |$\\beta=2$ |0.7844|
> |$\\beta=3$ |0.7824|
> |$\\beta=4$ |0.7841|
> |$\\beta=5$ |0.7830|
>
> The results reveal that $\\beta$ values within $\\{0,\\cdots,5\\}$ yield comparable results, with $\\beta=0$ achieving the best performance. This is attributed to the test AUC metric being primarily focused on the accuracy of $y$, rather than the whole generative model $p(x,r,y,z,c)$.
>
> We hope these clarifications address your concerns and demonstrate the validity and applicability of our proposed method. Thank you again for your valuable feedback.

---

### Decision · Program_Chairs · 2025-05-01

**Decision:**

Accept (poster)

**Comment:**

This paper studies a regression model where the response variable is stochastically monotonic with respect to one ($r$) of the two covariates ($r$ and $x$). They first reduce this monotonic regression model to a monotonic classification model and then show a simple yet interesting and quite powerful result that states that the monotonic assumption is equivalent to the existence of a latent variable that is independent of $r$ conditionally on $x$. This result allows them to learn the model without introducing regularization techniques, which are costly and introduce a possibly large bias in practice. They adopt a generative approach to model and learn this latent variable using variational inference. Numerical experiments suggest that their method outperforms traditional ones.

Two reviewers are positive/very positive about this work, both for its clarity and its content, which also provides good intuition and smart ideas. They raise important questions related to the assumptions that may be quite restrictive (e.g., boundedness of the covariate $r$), which were discussed in the rebuttal.

One reviewer was very negative, but more on the general idea of using generative modeling rather than on the paper itself. Therefore, it does not outweigh the other positive reviews and I recommend accepting this paper. Please, note that the rebuttal discussions should be included in the final version of the paper.